# Loss of *Asxl2* leads to myeloid malignancies in mice

Jianping Li[1,*], Fuhong He[2,*], Peng Zhang[1,*], Shi Chen[1], Hui Shi[1,3], Yanling Sun[2,4], Ying Guo[1], Hui Yang[1], Na Man[1], Sarah Greenblatt[1], Zhaomin Li[1], Zhengyu Guo[5], Yuan Zhou[3], Lan Wang[1], Lluis Morey[1], Sion Williams[1], Xi Chen[1,6], Qun-Tian Wang[7], Stephen D. Nimer[1], Peng Yu[5], Qian-Fei Wang[2,4], Mingjiang Xu[1] & Feng-Chun Yang[1]

*ASXL2* is frequently mutated in acute myeloid leukaemia patients with *t*(8;21). However, the roles of *ASXL2* in normal haematopoiesis and the pathogenesis of myeloid malignances remain unknown. Here we show that deletion of *Asxl2* in mice leads to the development of myelodysplastic syndrome (MDS)-like disease. *Asxl2*$^{-/-}$ mice have an increased bone marrow (BM) long-term haematopoietic stem cells (HSCs) and granulocyte–macrophage progenitors compared with wild-type controls. Recipients transplanted with *Asxl2*$^{-/-}$ and *Asxl2*$^{+/-}$ BM cells have shortened lifespan due to the development of MDS-like disease or myeloid leukaemia. Paired daughter cell assays demonstrate that *Asxl2* loss enhances the self-renewal of HSCs. Deletion of *Asxl2* alters the expression of genes critical for HSC self-renewal, differentiation and apoptosis in Lin$^-$cKit$^+$ cells. The altered gene expression is associated with dysregulated H3K27ac and H3K4me1/2. Our study demonstrates that ASXL2 functions as a tumour suppressor to maintain normal HSC function.

[1] Sylvester Comprehensive Cancer Center, Department of Biochemistry and Molecular Biology, University of Miami Miller School of Medicine, 1011 NW 15th Street, Room 417, Miami, Florida 33136, USA. [2] Key Laboratory of Genomic and Precision Medicine, Collaborative Innovation Center of Genetics and Development, Beijing Institute of Genomics, Chinese Academy of Sciences, Beijing 100101, China. [3] State Key Laboratory of Experimental Hematology, Institute of Hematology and Blood Diseases Hospital, Chinese Academy of Medical Sciences and Peking Union Medical College, Tianjin 300020, China. [4] University of Chinese Academy of Sciences, Beijing 100049, China. [5] Department of Electrical and Computer Engineering, and TEES-AgriLife Center for Bioinformatics and Genomic Systems Engineering, Texas A&M University, College Station, Texas 77843, USA. [6] Department of Public Health Sciences, University of Miami Miller School of Medicine, Miami, Florida 33136, USA. [7] Department of Biological Sciences, University of Illinois at Chicago, Chicago, Illinois 60607, USA. * These authors contributed equally to this work. Correspondence and requests for materials should be addressed to Q.-F.W. (email: wangqf@big.ac.cn) or to M.X. (email: mxx51@miami.edu) or to F.-C.Y. (email: fxy37@med.miami.edu).

Haematopoietic stem cells (HSCs) have the ability to self-renew and give rise to all lineages of blood cells. The balance between HSC self-renewal and lineage specification is governed by the interplay between external and internal stimuli. Epigenetic factors have been shown to function as regulators for haematopoiesis. Epigenetic alterations are frequently associated with myeloid malignancies by establishing specific gene expression profiles[1]. Polycomb group (PcG) proteins are prominent epigenetic regulators and form multi-subunit complexes, which modify chromatin to maintain or repress the expression of their target genes[2]. ASXL1, 2 and 3 that comprise the additional sex comb-like family are putative PcG proteins, which share a common domain architecture of conserved amino-terminal ASXN, ASXH domains and a carboxy-terminal plant homeodomain (plant homeodomain finger)[3]. As a chromatin regulator, ASXL1 has been shown to exert its function by recruiting the polycomb repressive complex 2, which adds methyl mark(s) on histone H3 lysine 27 (H3K27me1, H3K27me2 and H3K27me3)[4–6]. *Asxl1* loss leads to the development of myeloid malignancies in mice, which is associated with dysregulation of H3K27me3 (refs 6,7). Lai *et al.*[8] previously reported that *ASXL2* regulates gene expression in the heart at cell-type and stage-specific manners. Enhancers and other distal regulatory elements are critical for context-specific gene regulation. Such elements are associated with characteristic chromatin marks, including H3K4me1 and H3K27ac, which facilitate their identification.

Somatic mutations and chromosomal translocations are major drivers of a range of haematopoietic malignancies. Genomic studies have revealed that *ASXL1* and *ASXL2* are frequently mutated in myeloid malignancies[9–11]. *ASXL1* mutations occur in multiple spectra of myeloid malignancies[9,12]. In contrast, *ASXL2* mutations are largely restricted to patients with t(8;21) acute myeloid leukaemia (AML), both in children and in adults at an overall incidence of ~23% (refs 13,14). Interestingly, *ASXL1* and *ASXL2* mutations are mutually exclusive in t(8;21) AML patients[13], suggesting a common mechanism in promoting myeloid malignancies. Despite intense investigation, the role of *ASXL2* in normal and malignant haematopoiesis remains unknown.

In the current study, utilizing a murine model of *Asxl2* loss, we sought to explore the role of ASXL2 in normal and malignant haematopoiesis, and to identify the mechanisms by which *Asxl2* loss contribute to myeloid malignancies. We showed that *Asxl2* loss led to a myelodysplastic syndrome (MDS)-like disease in mice as evidenced by pancytopenia, dysplastic features of myeloid cells and skewed differentiation towards myeloid lineages. *Asxl2*$^{-/-}$ mice had increased frequency of long-term (LT) HSCs. Transplantation of bone marrow (BM) cells from *Asxl2*$^{-/-}$ or *Asxl2*$^{+/-}$ mice led to a leukaemic transformation in the lethally irradiated recipients, indicating a cell-autonomous effect of *Asxl2* loss on the pathogenesis of myeloid malignancies. Convergent analyses of RNA sequencing (RNA-seq) and chromatin immunoprecipitation (ChIP) assays followed by sequencing (ChIP-seq) data in BM Lin$^-$cKit$^+$ (LK) cells identified a subset of differentially expressed genes (DEGs), enriched with critical genes for HSC function and apoptosis, as well as myelopoiesis. Importantly, the altered gene expression was associated with dysregulated histone marks, including H3K27ac and H3K4me1/2. Our results demonstrate a critical role of ASXL2 in the maintenance of normal HSC functions.

## Results

***Asxl2* is required for normal haematopoiesis.** To determine *Asxl2* expression in wild-type (WT) haematopoietic lineages, different haematopoietic cell subpopulations were used for quantitative real-time PCR (qPCR). *Asxl2* was ubiquitously expressed in all cell subpopulations/lineages examined, including Lin$^-$Sca1$^+$cKit$^+$ (LSK), common myeloid progenitor, granulocyte–monocyte progenitor (GMP), megakaryocyte–erythrocyte progenitor, neutrophils (NEs), monocytes, erythrocytes (E), megakaryocytes, B cells and T cells (Supplementary Fig. 1a).

To elucidate the effects of *Asxl2* on haematopoiesis, we performed a series of haematopoietic phenotypic analyses using 6–8-week-old *Asxl2*$^{+/-}$ and *Asxl2*$^{-/-}$ mice. The *Asxl2*-deficient mice were generated previously by inserting a gene trap cassette into the first intron of *Asxl2*, which disrupts ASXL2 expression[15]. Successful deletion of *Asxl2* in haematopoietic cells was shown by PCR, qPCR and western blotting, respectively (Supplementary Fig. 1b–d). Of note, *Asxl2* deletion did not affect the *Asxl1* messenger RNA level in *Asxl2*$^{-/-}$ BM cells (Supplementary Fig. 1e).

Peripheral blood (PB) counts revealed pancytopenia in *Asxl2*$^{-/-}$ mice (Fig. 1a). However, the proportion of NEs was significantly higher in the PB of *Asxl2*$^{-/-}$ mice compared with WT controls (Fig. 1a). May–Grünwald–Giemsa-stained PB smears of *Asxl2*$^{+/-}$ and *Asxl2*$^{-/-}$ mice showed dysplastic features, including hyposegmented (bilobed) NEs with fine nuclear bridging (consistent with pseudo Pelger–Huët), asymmetric nuclear structure and increased number of poly-chromatophilic red blood cells and Howell–Jolly bodies in erythrocytes (Fig. 1b). Analyses of BM histologic sections of *Asxl2*$^{+/-}$ and *Asxl2*$^{-/-}$ mice revealed an increase in the proportion of myeloid cells and relatively decreased erythroid islands compared with WT controls (Fig. 1c). Consistently, *Asxl2*$^{-/-}$ mice had an increased myeloid/erythroid ratio compared with WT mice (WT: $0.92 \pm 0.21$ versus *Asxl2*$^{-/-}$: $2.20 \pm 1.39$). Dysplastic features were also frequently seen in *Asxl2*$^{+/-}$ and *Asxl2*$^{-/-}$ BM sections, and cellular cytospins, including abnormal nuclear and cytoplasm proportion, nuclear budding and fragmentocytes (Fig. 1c). In addition, the BM cellularity in *Asxl2*$^{-/-}$ femur was significantly lower than that in WT controls (Supplementary Fig. 1f).

Despite the lower body weight, *Asxl2*$^{-/-}$ mice exhibited splenomegaly with a higher cellularity compared with WT mice (Fig. 1d and Supplementary Fig. 1g). The histologic analysis of the *Asxl2*$^{+/-}$ and *Asxl2*$^{-/-}$ spleen sections showed disrupted splenic architecture with an increased proportion of myeloid cells (Fig. 1e). These data suggest that ASXL2 is required for normal haematopoiesis and loss of *Asxl2* in mice leads to haematological characteristics resembling an MDS-like disease.

**ASXL2 is required for the maintenance of GMP and LT-HSCs.** To further characterize the *Asxl2*-loss-mediated dysregulation of haematopoiesis, flow cytometric analyses were performed on PB, BM and spleen cells of WT, *Asxl2*$^{+/-}$ and *Asxl2*$^{-/-}$ mice. An increased proportion of granulocytic/monocytic cells (Gr1$^+$/Mac1$^+$) was observed in the PB, BM and spleens of *Asxl2*$^{-/-}$ mice compared with WT mice (Fig. 2a,b and Supplementary Fig. 2a,b). Immunohistochemical staining showed massive accumulation of myeloperoxidase (MPO)-positive cells in the spleens of *Asxl2*$^{+/-}$ and *Asxl2*$^{-/-}$ mice compared with WT spleens (Supplementary Fig. 2c,d), verifying the increased proportion of myeloid cells in the *Asxl2*$^{+/-}$ and *Asxl2*$^{-/-}$ spleens. In contrast, a decreased proportion of the CD71$^+$/Ter119$^+$ population was observed in the BM of *Asxl2*$^{-/-}$ mice compared with WT controls (Supplementary Fig. 2e), suggesting a role of ASXL2 in erythropoiesis *in vivo*.

As the frequency and number of lymphocytes were significantly lower in the PB of *Asxl2*$^{-/-}$ mice compared with WT mice, we next performed flow cytometric analysis to further determine whether *Asxl2* deletion impairs lymphocyte development. Lower

frequencies of LMPP (lymphoid-primed multipotent progenitor) and CLP (common lymphoid progenitor) were observed in the BM of $Asxl2^{-/-}$ mice compared with WT and $Asxl2^{+/-}$ mice (Supplementary Fig. 3a). In addition, lower percentages of pre-B cells ($cKit^+/B220^+$ and $CD25^-/IgM^+$) were observed in $Asxl2^{-/-}$ BM cells (Supplementary Fig. 3b). A higher proportion of DN1 ($CD44^+/CD25^-/CD4^-/CD8^-$) and a lower percentage of $CD4^+/CD8^+$ cell population were observed in the thymus of

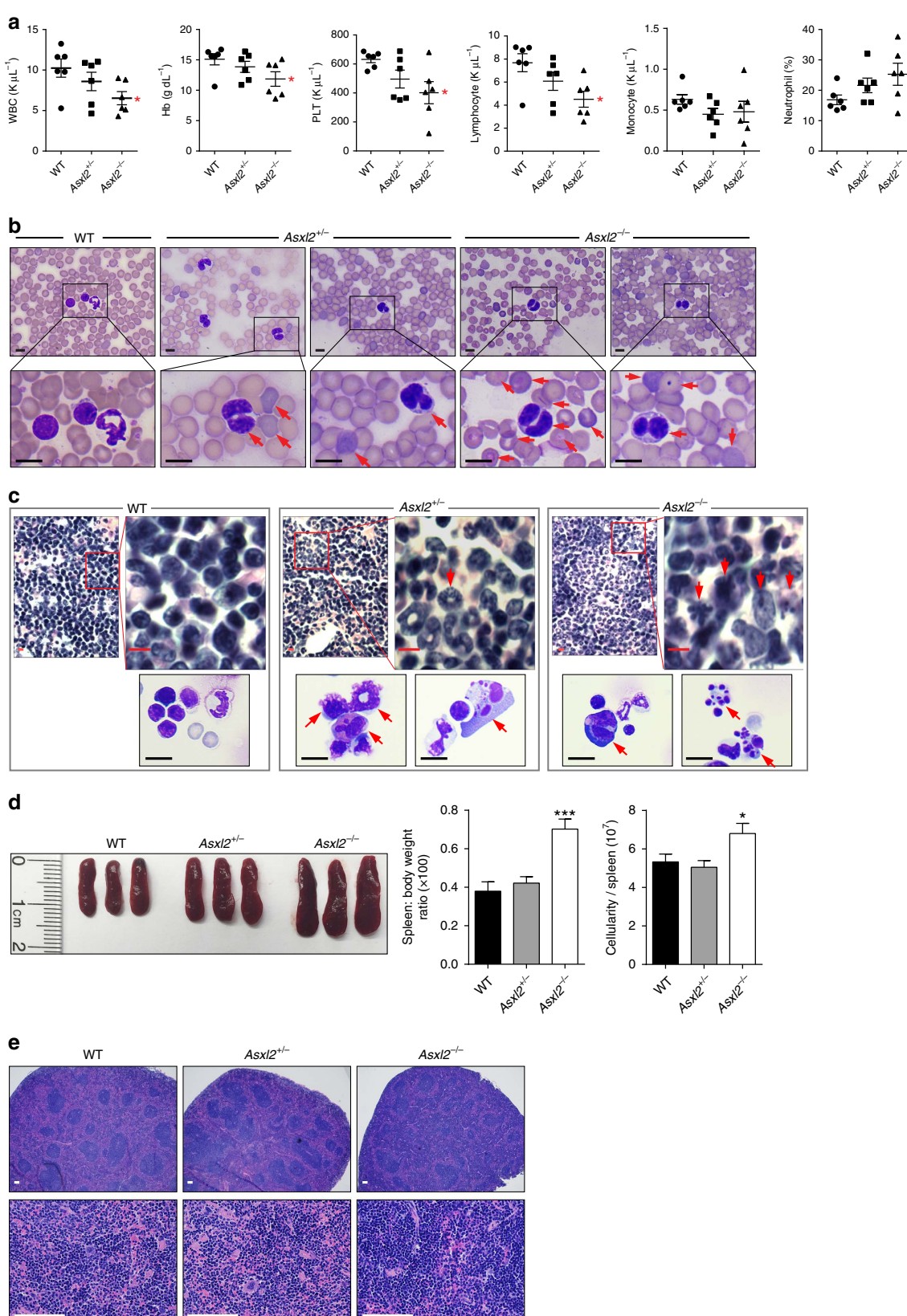

$Asxl2^{-/-}$ mice compared with WT and $Asxl2^{+/-}$ mice (Supplementary Fig. 3c). These data suggest that deletion of $Asxl2$ impairs the development of lymphocytes.

To assess whether $Asxl2$ deletion affects the pool of haematopoietic stem/progenitor cells (HSC/HPCs), we performed phenotypic analysis on the BM cells of WT, $Asxl2^{+/-}$ and $Asxl2^{-/-}$ mice by flow cytometry. $Asxl2^{-/-}$ BM cells had a significantly higher proportion of LSK cells compared with WT BM cells (Fig. 2c,d). The $Lin^-cKit^+Sca1^-$ ($LKS^-$) cell populations in the BM of $Asxl2^{-/-}$ mice were similar to that of WT mice (Supplementary Fig. 3d). Further analysis showed that deletion of $Asxl2$ significantly increased the LT-HSCs ($LSKCD34^-FLK2^-$ cells) and short-term-HSCs ($LSKCD34^+$ $FLK2^-$ cells) in the BM compared with WT and $Asxl2^{+/-}$ mice (Fig. 2e,f). The percentage of multipotential progenitor cells was similar in the BM of $Asxl2^{-/-}$, WT and $Asxl2^{+/-}$ mice (Supplementary Fig. 3d). When each myeloid progenitor population was analysed within the $LKS^-$ populations, the percentage of GMP populations were significantly higher, whereas megakaryocyte–erythrocyte progenitor and common myeloid progenitor populations were slightly lower in the BM of $Asxl2^{-/-}$ mice compared with WT and $Asxl2^{+/-}$ mice (Fig. 2g,h and Supplementary Fig. 3d). Will et al.[16] reported that higher-risk MDS have a higher proportion of GMP. The higher proportion of GMP in $Asxl2$-deficient mice may associate with disease progression in vivo.

Interestingly, aged $Asxl2^{-/-}$ mice (around 12 months of age) had significantly higher white blood cell and NE counts in the PB than WT and $Asxl2^{+/-}$ mice (Supplementary Fig. 4a). These aged $Asxl2^{-/-}$ mice also exhibited weight loss, splenomegaly and pale femurs (Supplementary Fig. 4b). Flow cytometric analysis showed an increased proportion of $Gr1^+/Mac1^+$ cells (Supplementary Fig. 4c), higher percentage of LT-HSCs and GMPs (Supplementary Fig. 4d,e) in their BM.

Fetal liver (FL) is an important haematopoietic site in mice. Flow cytometric analysis on the FL cells of WT, $Asxl2^{+/-}$ and $Asxl2^{-/-}$ embryos (day 12.5) mice revealed increased multipotential progenitor cell and $Gr1^+/Mac1^+$ cell populations in $Asxl2^{-/-}$ FL cells, whereas the frequencies of LSK and GMP were comparable among the three genotypes (Supplementary Fig. 5a). Consistently, the frequencies of colony-forming unit (CFU)- mixed colonies of granulocytes/macrophage (GM), E and megakaryocytic cells, CFU-GM and burst forming unit-E were significantly higher in $Asxl2^{-/-}$ FL cells than in WT and $Asxl2^{+/-}$ cells (Supplementary Fig. 5b).

Collectively, these data suggest that ASXL2 is required for the maintenance of normal HSC/HPC pools and lineage commitment in vivo.

**Deletion of $Asxl2$ increases HSC self-renewal capacity.** To determine the effect of $Asxl2$ loss on the function of HSC/HPCs, we next assayed the frequency of CFU cells (CFU-C) in the BM and spleens of WT, $Asxl2^{+/-}$ and $Asxl2^{-/-}$ mice. The frequencies of CFU-Cs, CFU-GM and CFU-mixed colonies of GM, E and megakaryocytic cells were significantly higher in the BM and spleens of $Asxl2^{-/-}$ mice (Fig. 3a and Supplementary Fig. 6a), suggesting a preferential myeloid progenitor increment. To determine the proliferative potential of multipotent progenitor cells, we assayed high proliferative potential colony-forming cells (HPP-CFC) using double-layer agar cultures[17]. The frequencies of HPP-CFC were significantly higher in the BM of $Asxl2^{+/-}$ and $Asxl2^{-/-}$ mice compared with WT mice (Fig. 3b).

Balanced symmetric and asymmetric cell division is required for the preservation of normal stem cell pool and production of blood cells. To further determine whether loss of $Asxl2$ alters cell fates of HSC/HPCs, paired daughter cell assays[18] were performed to assess the proportions of symmetric self-renewal, symmetric division and asymmetric division using $CD34^-LSK$ cells isolated from WT and $Asxl2^{-/-}$ BM. $Asxl2$ loss resulted in a higher proportion of cells with symmetric self-renewal capacity (62%) compared with WT cells (33%, Fig. 3c). In contrast, the cells with symmetric differentiation potential were significantly reduced in $Asxl2^{-/-}$ $CD34^-LSK$ cells (18%) compared with WT cells (40%), whereas the frequency of cells that underwent asymmetric differentiation in $Asxl2^{-/-}$ $CD34^-LSK$ cells was comparable to that of WT cells (Fig. 3c). To determine the replating potential of $CD34^-LSK$ cells over time, we replated the cells that underwent symmetric self-renewal. Although a steady decline of replating potential was observed in WT cells, a significantly higher replating potential was observed over three successive replating times in $Asxl2^{-/-}$ cells (Fig. 3d). In addition, the progeny of the single cell culture showed that $Asxl2^{-/-}$ $CD34^-LSK$ cells preferentially gave rise to larger proportions of granulocytic/ monocytic cells, whereas the WT $CD34^-LSK$ cells had the potential to differentiate into diverse cell lineages, including erythroid cells and myeloid cells (Fig. 3e). These data indicate that ASXL2 is critical for maintaining the balance of symmetric and asymmetric divisions in HSCs.

The changes in the HSC/HPCs frequencies in vivo can be associated with altered cell proliferation and/or apoptosis. We then examined whether $Asxl2$ loss affected cell survival of HSC/HPCs by flow cytometric analysis on BM cells following Annexin V and 7-amino-actinomycin D (7-AAD) staining. $Asxl2^{-/-}$ LK cells had a significantly lower proportion of Annexin $V^+$ cells than WT cells (Fig. 3f,g). In contrast, a higher percentage of Annexin $V^+$ cells was observed in matured cells ($Lin^+$) of $Asxl2^{-/-}$ BM than that of WT BM (Supplementary Fig. 6b,c). These data indicate that deletion of $Asxl2$ led to a reduced apoptosis of HSC/HPCs, but an increased apoptosis in more differentiated BM cells. Furthermore, 5-bromodeoxyuridine (BrdU) incorporation assays revealed an increased frequency of $Asxl2^{+/-}$ and $Asxl2^{-/-}$ LK cells at S-phase than WT LK cells, suggesting a higher proliferation of $Asxl2^{+/-}$ and $Asxl2^{-/-}$ LK cells (Supplementary Fig. 6d). These results demonstrate that loss of $Asxl2$ altered the apoptosis and proliferation of LK cells.

**Figure 1 | Development of an MDS-like disease in $Asxl2^{-/-}$ mice.** (**a**) Parameters of PB were summarized from WT, $Asxl2^{+/-}$ and $Asxl2^{-/-}$ mice at 6–8 weeks of age (six mice per genotype). (**b**) May–Grünwald–Giemsa stained PB smears from representative WT, $Asxl2^{+/-}$ and $Asxl2^{-/-}$ mice were shown. The PB smears of $Asxl2^{+/-}$ and $Asxl2^{-/-}$ mice showed dysplastic features, including hyposegmented (bilobed) NEs with fine nuclear bridging consistent with pseudo Pelger–Huët, asymmetric nuclear structure and increased number of polychromatophilic red blood cells (RBCs) and Howell–Jolly bodies in erythrocytes. Scale bar, 10 μm. (**c**) Haematoxylin and eosin (H&E) stained sections of femurs (upper panels) and May–Grünwald–Giemsa stained cytospins (lower panels) from representative WT, $Asxl2^{+/-}$ and $Asxl2^{-/-}$ mice (6–8 week old). Scale bar, 10 μm. (**d**) Enlarged spleens in $Asxl2^{-/-}$ mice. Shown are representative photographs of spleens of three sets of WT, $Asxl2^{+/-}$ and $Asxl2^{-/-}$ mice (6–8 week old), spleen/body weight ratio ($\times 100$) and spleen cellularity for each genotype of mice (six mice per genotype). (**e**) H&E staining of paraffin-embedded sections of spleens from representative WT, $Asxl2^{+/-}$ and $Asxl2^{-/-}$ mice (6–8 week old). Red arrows indicate dysplastic cells. Scale bar, 100 μm. Data are presented as mean ± s.e.m. Comparisons among the groups were performed by one-way analysis of variance. $*P < 0.05$ and $***P < 0.001$.

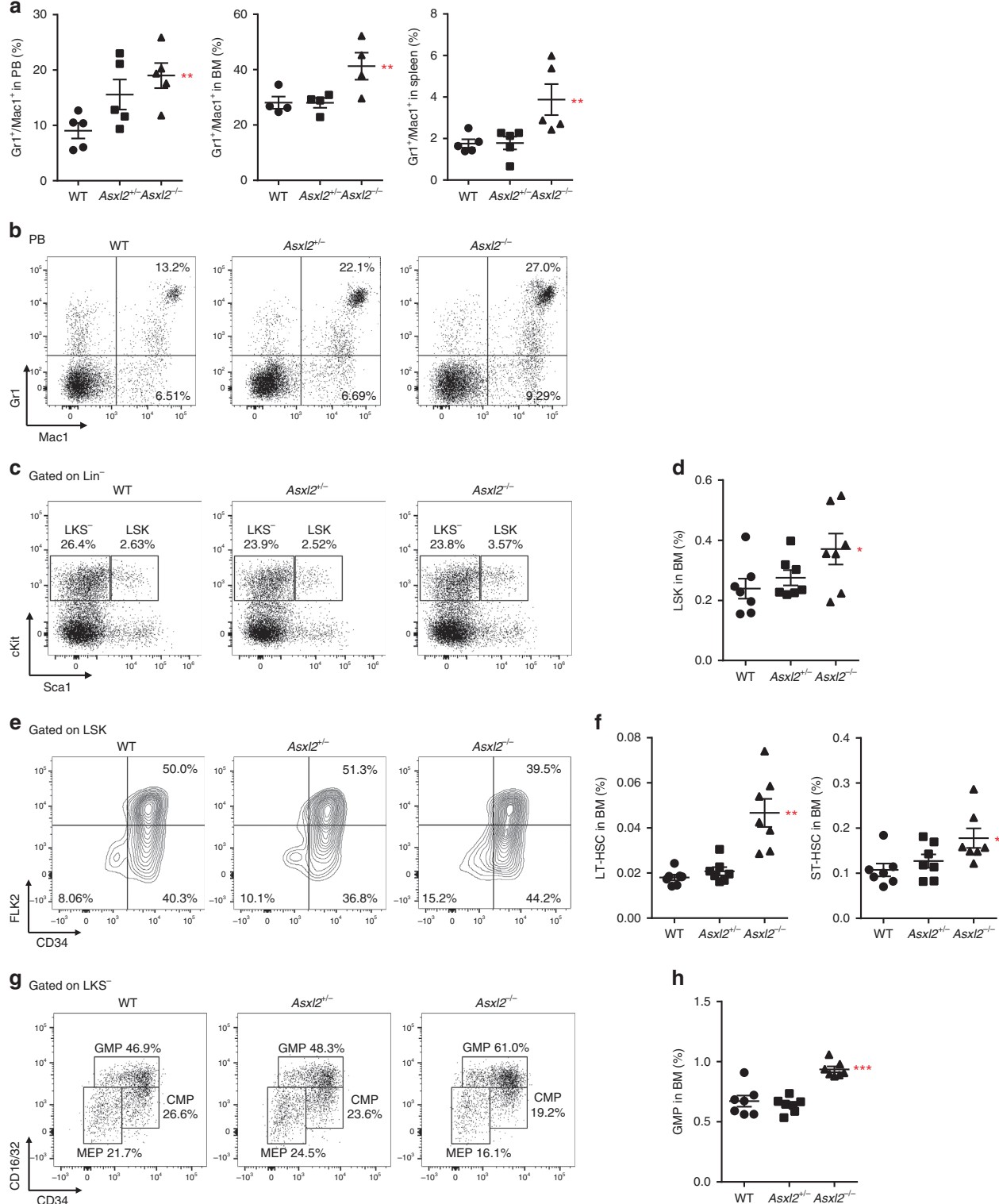

**Figure 2 | Altered HSC and myeloid progenitor cell populations in *Asxl2*$^{-/-}$ mice. (a)** Quantification of the per cent Gr1$^+$/Mac1$^+$ cell populations in the PB, BM and spleen of WT, *Asxl2*$^{+/-}$ and *Asxl2*$^{-/-}$ mice (6–8 week old, four to five mice per genotype). **(b)** Flow cytometric analysis of Gr1$^+$/Mac1$^+$ cell populations in PB of representative WT, *Asxl2*$^{+/-}$ and *Asxl2*$^{-/-}$ mice. **(c)** Flow cytometric analysis of Lin$^-$cKit$^+$Sca1$^+$ (LSK) and Lin$^-$cKit$^+$Sca1$^-$ (LKS$^-$) compartments in the BM of representative WT, *Asxl2*$^{+/-}$ and *Asxl2*$^{-/-}$ mice. **(d)** Quantification of the per cent LSK population in the total BM cells of each genotype of mice (6–8 week old, seven mice per genotype). **(e)** Flow cytometric analysis of LT-HSC, short-term (ST)-HSC and MPP compartments in BM LSK cells of representative WT, *Asxl2*$^{+/-}$ and *Asxl2*$^{-/-}$ mice. **(f)** Quantification of the per cent LT-HSC and ST-HSC populations in the total BM cells of each genotype of mice (seven mice per genotype). **(g)** Flow cytometric analysis of common myeloid progenitor (CMP), GMP and megakaryocyte–erythrocyte progenitor (MEP) cell populations in BM LKS$^-$ cells of representative WT, *Asxl2*$^{+/-}$ and *Asxl2*$^{-/-}$ mice. **(h)** Quantitative analysis of GMP populations in the total BM cells of each genotype of mice (6–8 week old, seven mice per genotype). Data are presented as mean ± s.e.m. Comparisons among the groups were performed by one-way analysis of variance. *$P < 0.05$, **$P < 0.01$ and ***$P < 0.001$.

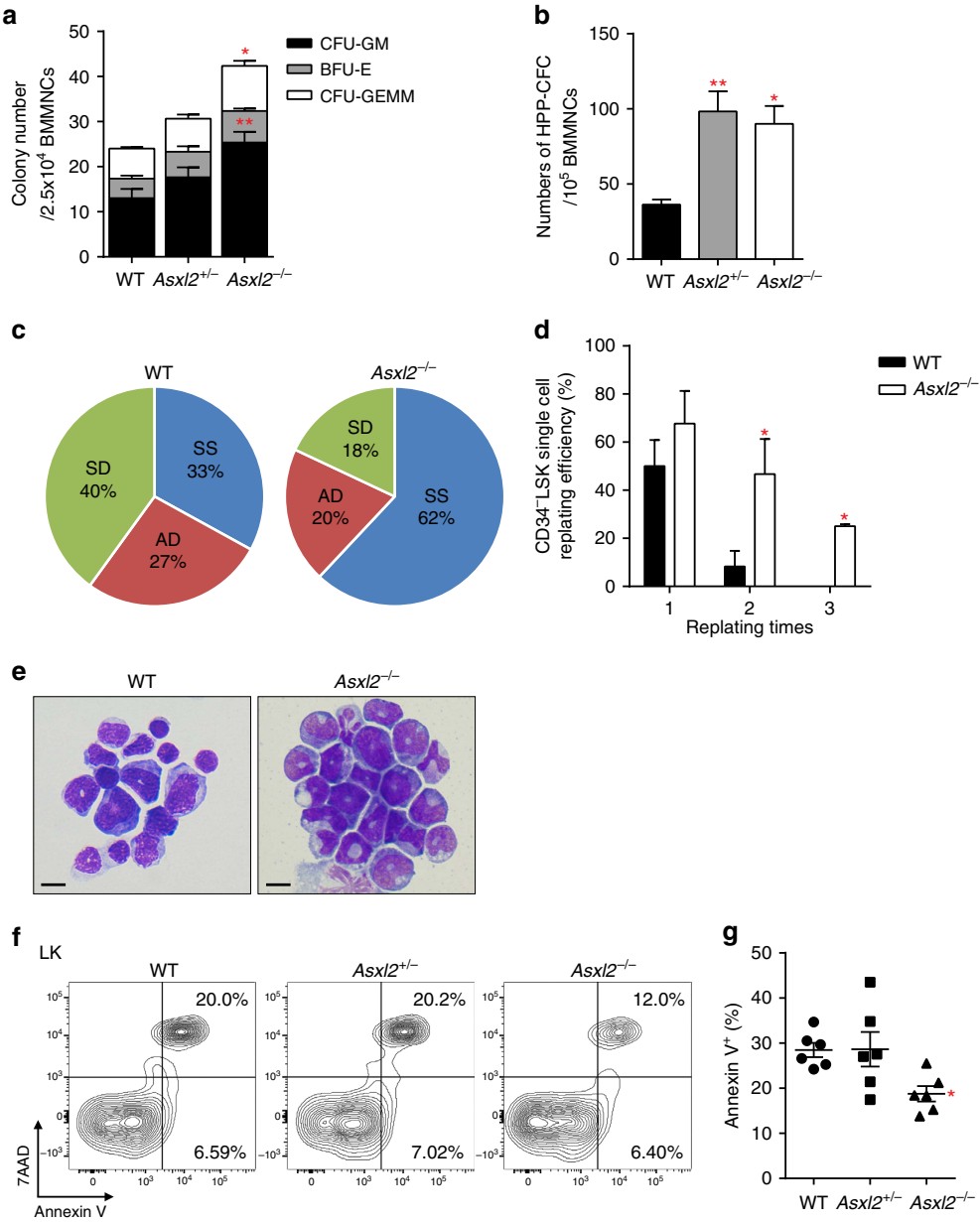

**Figure 3 | Loss of *Asxl2* alters HSC self-renewal and cell fates.** (**a**) CFUs in BM cells from WT, *Asxl2*[+/−] and *Asxl2*[−/−] mice (6–8 week old, three mice per genotype) were assessed in semi-solid media in the presence of mouse (m) SCF, mIL-3, human (h) IL-6, mTPO, mouse granulocyte–macrophage colony-stimulating factor (mGM-CSF) and mEPO. BFU-E, burst forming unit-erythrocyte; GEMM, mixed colonies of GM, E, and megakaryocytic cells; GM, granulocytes/macrophages. BMMNCs, bone marrow mononuclear cells. (**b**) HPP-CFC were determined in the BM of WT, *Asxl2*[+/−] and *Asxl2*[−/−] mice (6–8 week old, three mice per genotype) in the presence of mSCF, human IL6 (hIL-6), hFlt3L, mIL-3, mGM-CSF and mG-CSF. The quantification of large colonies (≥1 mm) was enumerated on the 14th day of cultures. (**c**) Paired daughter cell assays were performed at single cell level on CD34[−]LSK cells isolated from WT and *Asxl2*[−/−] mice, and each analysed for symmetric self-renewal (SS, blue), asymmetric cell division (AD, red) or symmetric differentiation (SD, green). (**d**) Single-cell replating assays on the CD34[−]LSK cells isolated from WT and *Asxl2*[−/−] mice were performed to determine replating capacity (6–8 week old, three mice per genotype). (**e**) Representative images of May–Grünwald–Giemsa-stained cytospin preparations of cells derived from WT and *Asxl2*[−/−] CD34[−]LSK cell cultures. Scale bar, 10 μm. (**f**) Apoptosis analysis (Annexin V/7-AAD staining) on freshly isolated Lin[−]cKit[+] (LK) cells from BM of representative WT, *Asxl2*[+/−] and *Asxl2*[−/−] mice. (**g**) Quantification of the apoptotic cells (Annexin V[+]) within LK cells of BM from WT, *Asxl2*[+/−] and *Asxl2*[−/−] mice (6–8 week old, six mice per genotype). The colony assay was performed in three replicates. Data are presented as mean ± s.e.m. Comparisons among more than two groups were performed by one-way analysis of variance test. Comparisons between two experimental groups were performed by two-tailed unpaired Student's *t*-test. *$P < 0.05$ and **$P < 0.01$.

**Cell-autonomous effect of *Asxl2* loss in HSC/HPC functions.** To assess whether *Asxl2*-loss-mediated alteration in HSC/HPC function is cell autonomous, we next performed competitive transplantation assays to examine the repopulating capacity of *Asxl2*-deficient BM cells (Supplementary Fig. 7a). When the

donor cell chimerism was analysed kinetically in the PB of recipient mice, the CD45.2 cell population remained ∼50% in mice receiving WT and *Asxl2*[+/−] BM cells, whereas a transient decline in CD45.2[+] cells was observed in recipients transplanted with *Asxl2*[−/−] BM cells in the first 3 months. Interestingly, the

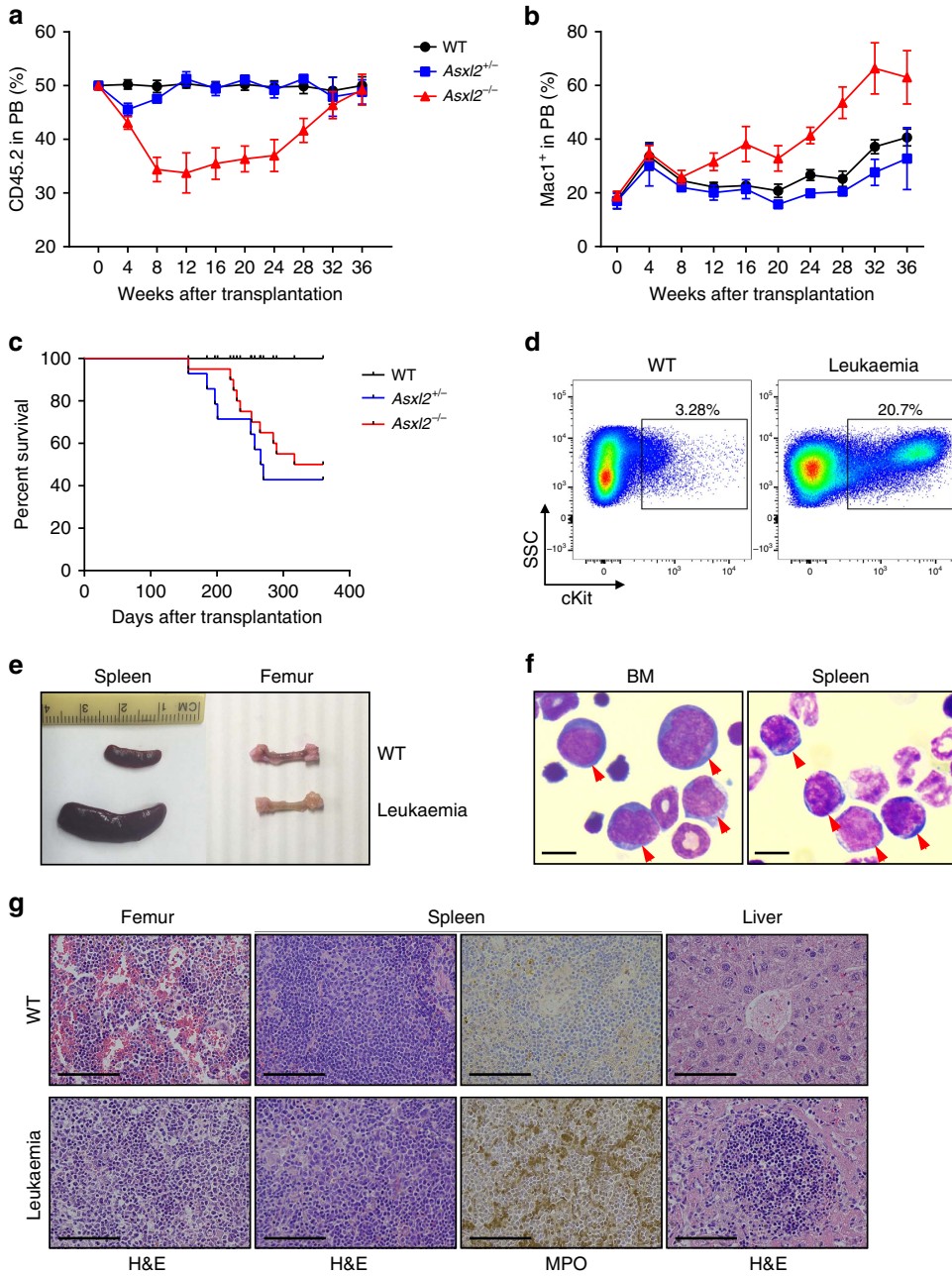

**Figure 4 | Cell-autonomous effect of malignant progression in *Asxl2*<sup>+/−</sup> and *Asxl2*<sup>−/−</sup> mice.** (**a**) The kinetics of CD45.2 chimerism in the PB of mice receiving WT, *Asxl2*$^{+/-}$ or *Asxl2*$^{-/-}$ BM cells (mean ± s.e.m., four to eight mice per genotype). (**b**) The kinetics of percent Mac1$^+$ cells in CD45.2$^+$ cells of mice receiving WT, *Asxl2*$^{+/-}$ or *Asxl2*$^{-/-}$ BM cells (mean ± s.e.m., four to eight mice per genotype). (**c**) Kaplan–Meier curve shows the per cent survival of mice received WT, *Asxl2*$^{+/-}$ or *Asxl2*$^{-/-}$ BM cells over time. (**d**) Flow cytometric analysis of cKit$^+$ cells in BM. (**e**) Representative gross appearance of spleens and bones. (**f**) Representative images of May–Grünwald–Giemsa-stained cytospin preparations of BM and spleen cells from representative leukemic mice receiving *Asxl2*$^{-/-}$ BM cells. Red arrows indicate myeloblastic cells. Scale bar, 10 μm. (**g**) H&E-stained sections of the femurs, spleens and livers, and MPO-stained spleen sections from a representative leukemic recipient mouse transplanted with *Asxl2*$^{-/-}$ BM cells and a recipient mouse receiving WT BM cells. Scale bar, 100 μm.

CD45.2 chimerism was steadily increased in the recipients transplanted with *Asxl2*$^{-/-}$ BM cells afterwards (Fig. 4a). Furthermore, *Asxl2*$^{-/-}$ BM cells contributed to a greater proportion of Mac1$^+$ myeloid cells in the PB than WT BM cells (Fig. 4b).

Five months after transplantation, mice receiving *Asxl2*$^{+/-}$ and *Asxl2*$^{-/-}$ BM cells started to become sick, whereas all the recipients transplanted with WT BM cells were healthy. These *Asxl2*$^{+/-}$ and *Asxl2*$^{-/-}$ BM recipient mice exhibited weight loss,

pale foot pads and lack of activity, indicative of cachexia symptoms (Supplementary Fig. 7b). These recipients had a greatly shortened lifespan compared with the WT BM recipients (Fig. 4c). Necropsies of the moribund/deceased mice revealed that a fraction of *Asxl2*$^{+/-}$ and *Asxl2*$^{-/-}$ BM recipients had splenomegaly, severe anemia, decreased platelet counts and significantly higher percentages ( >20%) of cKit$^+$ cells in the BM (Fig. 4d,e and Supplementary Fig. 7c,d). Myeloid blast cells were also seen in the BM, spleen and PB of these diseased mice (Fig. 4f and Supplementary Fig. 7e,

respectively). Histologic analyses showed infiltration of myeloid cells in the BM, spleen and liver (Fig. 4g). Furthermore, flow cytometric analysis revealed increased Mac1$^+$ population in the BM, spleen and PB, and higher percentages of cKit$^+$/Mac1$^+$ cells in the BM and spleen of these diseased mice (Supplementary

Fig. 7f). In contrast, lower percentages of erythroid cells and megakaryocytes were observed in the BM of these diseased recipients (Supplementary Fig. 7g,h). Collectively, these data indicate that a fraction of recipients transplanted with $Asxl2^{+/-}$ and $Asxl2^{-/-}$ BM cells developed myeloid leukaemia.

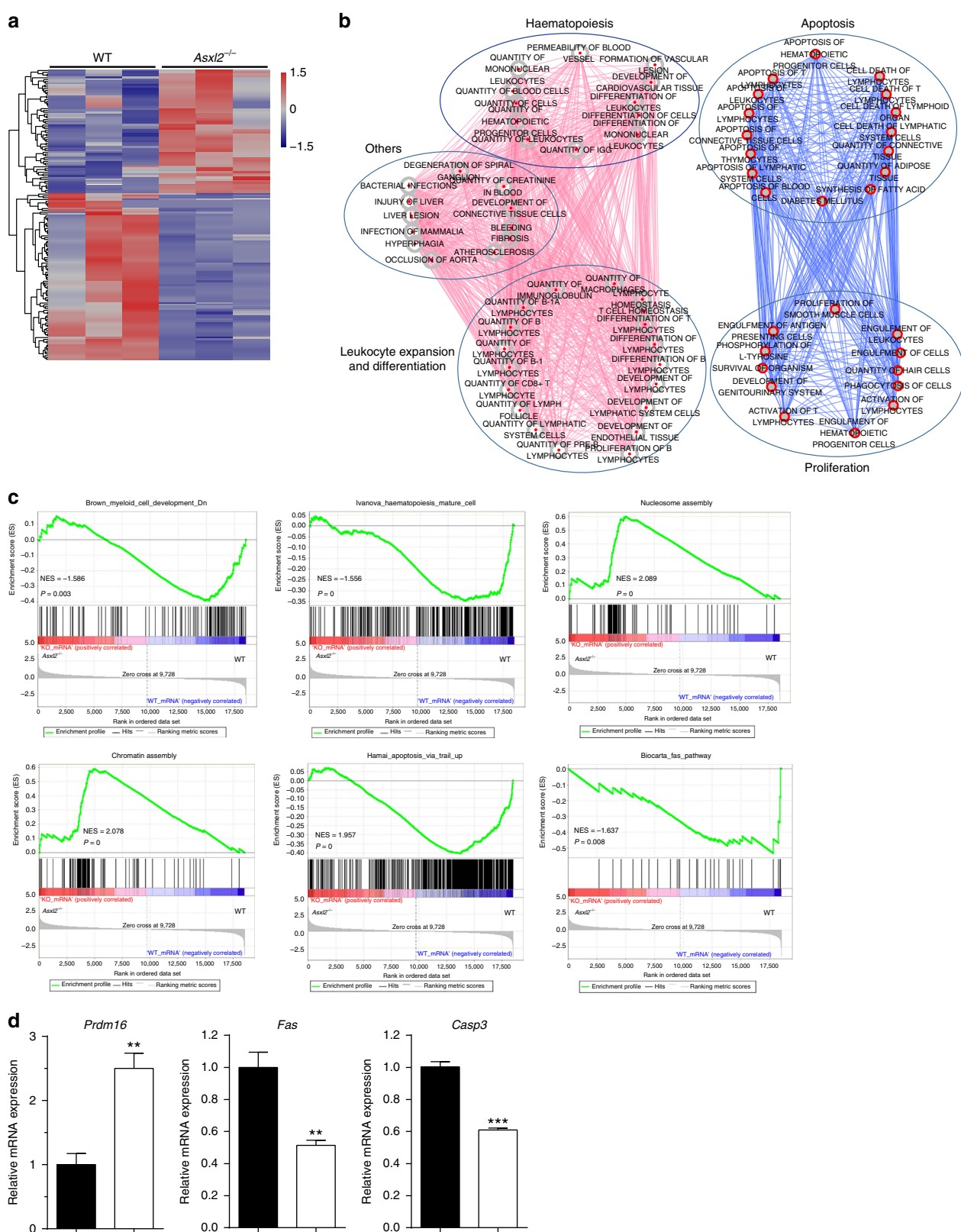

Of note, a fraction of the recipient mice received $Asxl2^{+/-}$ or $Asxl2^{-/-}$ BM cells developed the MDS-like disease as evidenced by dysplastic features of myeloid lineages, pancytopenia and myeloid infiltration in the spleen and liver (Supplementary Figs 7e and 8a,b, and Supplementary Table 3). In addition, a lower proportion of Annexin V$^+$ cells in BM LK cells and a higher percentage of Annexin V$^+$ cells in BM Lin$^+$ cells were noted in the diseased recipients (Supplementary Fig. 8c).

**ASXL2 is required for haematopoietic transcriptional program.** PcG proteins are a family of proteins responsible for transcriptional repression during cellular differentiation[19]. To delineate the effect of ASXL2 in the maintenance of HSC self-renewal and cell fates, we performed RNA-seq on LK cells isolated from 6-week-old WT and $Asxl2^{-/-}$ mice. Analysis of RNA-seq data revealed a set of 565 DEGs in $Asxl2^{-/-}$ LK cells (Supplementary Dataset 1, $P < 0.05$, a two-sided test for significance against a null hypothesis), in which 160 DEGs were shown (Fig. 5a, $P < 0.05$, false discovery rate (FDR) $< 0.25$ and fold change $\geq 1.5$). Functional enrichment analysis using Ingenuity Pathway Analysis (IPA) demonstrated that the DEGs were enriched for those related to haematopoiesis, apoptosis and cell differentiation (Fig. 5b and Supplementary Fig. 9a). Consistently, the gene set enrichment analysis (GSEA) showed that the genes with altered expression in $Asxl2^{-/-}$ LK cells were enriched for myeloid cell development, haematopoiesis, chromatin/nucleosome assembly, apoptosis and Fas pathway signatures (Fig. 5c).

$Asxl2^{-/-}$ LK cells had decreased mRNA expression levels of *Fas* and *Casp3*, but an increased level of *Prdm16* compared with WT cells (Fig. 5d). PRDM16 is required for normal haematopoiesis and highly expressed in a subgroup of AML patients[20]. To assess the effect of *Prdm16* overexpression on the haematopoietic phenotypes of $Asxl2^{-/-}$ FL cells, *Prdm16* was knocked down in $Asxl2^{-/-}$ FL cells (Supplementary Fig. 9b) and the frequencies of LSK and CFU-C were examined. *Prdm16*-KD significantly decreased the percentage of LSK cells, as well as the frequency of CFU-C in $Asxl2^{-/-}$ FL cells compared with WT cells (Supplementary Fig. 9c,d).

**ASXL2 maintains transcription through histone modifications.** It has been shown that deletion of *Asxl2* led to reduced global levels of H3K27me3 in heart cells[8], indicating that ASXL2 regulates H3K27 methylation. To determine whether *Asxl2* loss would have an impact on the global histone modification states in BM LK cells, we performed western blotting with related histone modification antibodies on WT and $Asxl2^{-/-}$ LK cells. Surprisingly, only a slight reduction of H3K4me3 and H3K27me3 levels was observed in $Asxl2^{-/-}$ cells (Supplementary Fig. 10a).

To explore the impact of ASXL2 loss on histone modifications in HSC/HPCs, we performed ChIP-seq using antibodies against H3K4me1, H3K4me2 and H3K27ac. Loss of *Asxl2* did not change the enrichment sites of each of the histone marks on the genomic regions (Supplementary Fig. 10b,c). Integrative analysis showed a significantly positive correlation between H3K27ac and gene expression in WT and $Asxl2^{-/-}$ LK cells, respectively ($R = 0.68$, $P < 0.001$; Supplementary Fig. 10d), which is consistent with previous studies[21,22]. Similarly, $R$-values ranging from 0.5 to 0.7 were identified for associations between H3K4me1/2 and gene expression in WT and $Asxl2^{-/-}$ LK cells, respectively ($P < 0.001$, Supplementary Fig. 10d). These data indicate that gene expression is positively associated with H3K27ac and H3K4me1/2.

We further confirmed that the changes in the three histone modifications were positively correlated with the change of mRNA expression ($R = \sim 0.25$, $P < 0.001$, Fig. 6a and Supplementary Fig. 10e). We identified 1,738, 1,106 and 1,267 regions with differential levels of enrichment between WT and $Asxl2^{-/-}$ LK cells for H3K27ac, H3K4me1 and H3K4me2, respectively. GSEA analysis revealed that genes with increased signals of H3K27ac, H3K4me1 or H3K4me2 were concordantly upregulated and genes with decreased signals of H3K27ac, H3K4me1 or H3K4me2 were concordantly downregulated on *Asxl2* deletion (Supplementary Fig. 11a).

In comparison with H3K4me1 and H3K4me2, the enrichment of H3K27ac had the most significant change at identified regions (Fig. 6b and Supplementary Fig. 11b). Furthermore, the genes with increased and/or decreased H3K27ac signals showed a greater degree of mRNA expression change compared with those with increased or decreased H3K4me1/2 signals, respectively (Fig. 6c and Supplementary Fig. 11c; paired $t$-test $P < 0.001$). Kyoto Encyclopedia of Genes and Genomes (KEGG) pathway analysis showed that genes with differential H3K27ac signals were enriched in signatures of haematopoietic cell lineage, cancer signalling pathway and myeloid leukaemia development (Fig. 6d). IPA analysis further confirmed that genes with altered enrichment levels of H3K27ac were enriched in myeloid cell differentiation and apoptosis pathways (Supplementary Fig. 12a–c). Of note, two dysregulated targets (*Prdm16* and *Fas*) were found to be associated with altered histone modifications (Fig. 6e). Meanwhile, two histone deacetylation-related proteins, HDAC1 and HDAC2, were identified to associate with ASXL2 in 293T cells by immunoprecipitation (IP) and western blotting (Supplementary Fig. 13a–c), implying the potential of ASXL2 involvement in the deacetylation of H3K27.

Our data indicate that alteration of H3K27ac enrichment had a greater impact on gene expression compared with H3K4me1/2 in *Asxl2*-deficient LK cells. Altogether, these data suggest that ASXL2 loss is associated with changes in gene expression through alterations in H3K27ac and H3K4me1/2. Further studies to elucidate the mechanism of ASXL2 in regulating HDAC activity are warranted.

## Discussion

Despite the clinical importance of *ASXL2* mutations in AML patients with $t(8;21)$, the role of ASXL2 in normal and malignant haematopoiesis remains unknown. *Asxl2* is expressed in HSCs and their differentiated HPCs and multiple

**Figure 5 | *Asxl2* deletion alters the expression of genes associated with myeloid cells and apoptosis in LK cells. (a)** The heatmap shows the cluster of 160 DEGs between WT and $Asxl2^{-/-}$ LK cells ($P < 0.05$, FDR $< 0.25$ and fold change $\geq 1.5$). Red and blue indicate 68 up- and 92 downregulated genes, respectively. **(b)** Enrichment map is used for visualization of the results of IPA analyses about Diseases and Biological Functions for 565 DEGs ($P < 0.05$). The network of enriched functional terms (Nodes, $P < 0.01$ and |z-score| $\geq 1.5$ in IPA) is shown. Edges indicate gene overlap between the enriched terms; thickness represents statistical significance of term enrichment. Only edges with a Fisher's exact test nominal $P < 10^{-4}$ were visualized. Terms linked by red lines with positive $z$-scores indicate that functional activity is increased, whereas that by blue ones with negative $z$-scores mean decreased activity. Oval indicates functional cluster, which includes relevant terms. **(c)** GSEA plots showing dysregulated gene expression of myeloid cell development, haematopoiesis, nucleosome/chromatin assembly, apoptosis and Fas pathway signatures in $Asxl2^{-/-}$ LK cells. The normalized enrichment score (NES) and $P$-value are shown. **(d)** qPCR showing upregulation of *Prdm16* and downregulation of *Casp3* and *Fas* in $Asxl2^{-/-}$ LK cells compared with WT cells. Data are presented as mean ± s.e.m. Comparisons between two groups were performed by two-tailed unpaired Student's $t$-test **$P < 0.01$ and ***$P < 0.001$.

haematopoietic lineages, suggesting that ASXL2 might play an important role in normal haematopoiesis. In this study, we demonstrated that ASXL2 is required for the normal function of HSCs/HPCs and deletion of *Asxl2* in mice increased LT-HSC pool and promoted differentiation towards granulocytic/mono-cytic lineages.

The majority of the AML patients with somatic *ASXL2* mutations are heterozygous and possess a WT allele and an *ASXL2* mutant allele, which leads to nonsense/frameshift[13], indicating that loss-of-function of ASXL2 might be important in the pathogenesis of myeloid malignancies. In the present study, we utilized an *Asxl2* knockout mouse model to show that

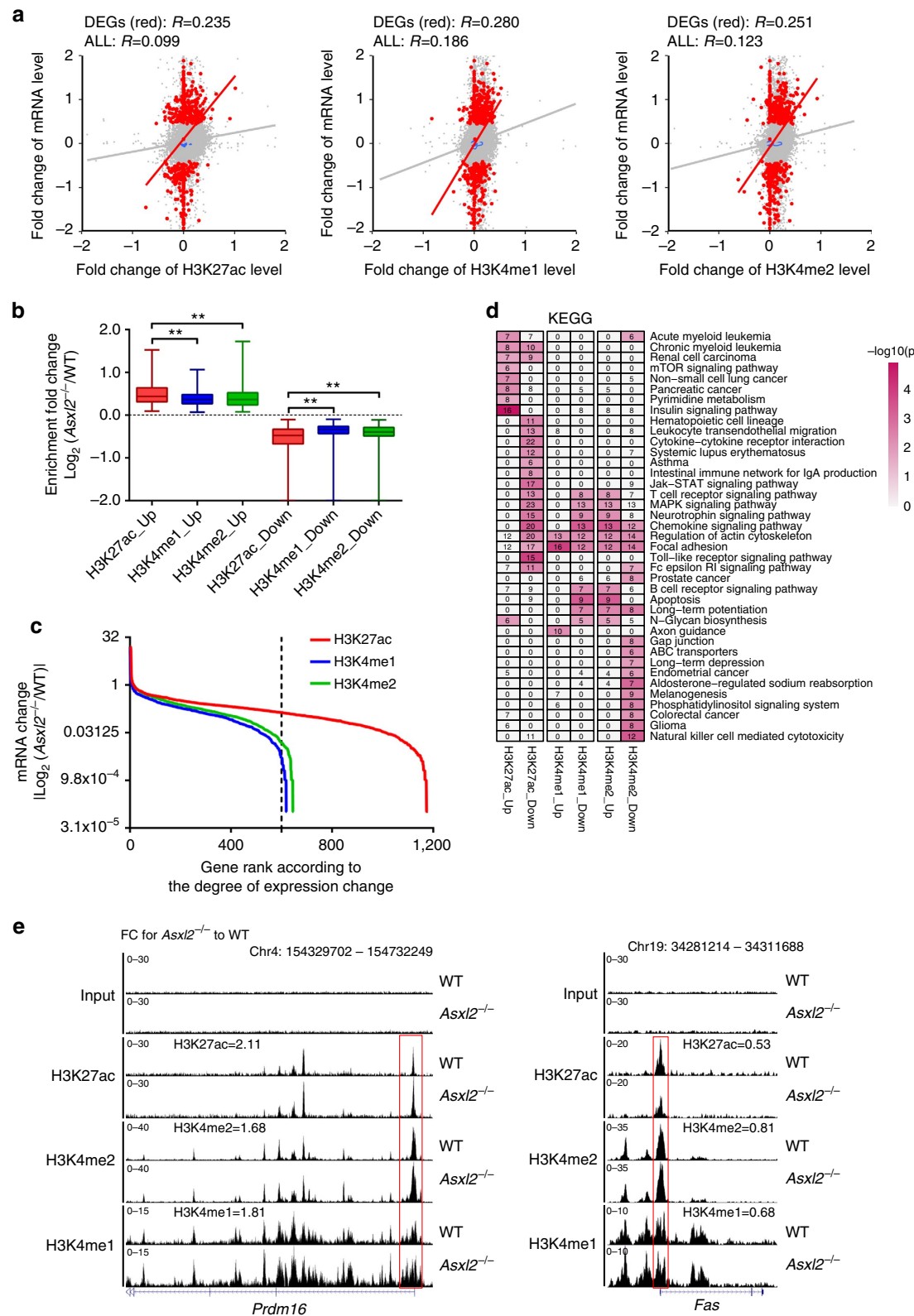

*Asxl2* loss in mice leads to dysplastic haematopoiesis, pancytopenia and splenomegaly, resembling an MDS-like disease. Therefore, ASXL2 functions as a tumour suppressor in myelopoiesis.

Normal HSCs are capable of asymmetric division, facilitating both self-renewal and the generation of differentiated progenies. Increased self-renewal potential is one of the hallmarks of cancer[23]. Our paired daughter cell assays revealed that deletion of *Asxl2* resulted in an increased symmetric division in HSCs, which probably facilitates the expansion of HSCs in *Asxl2*$^{-/-}$ mice. Indeed, there was a higher percentage of LT-HSCs in the BM of *Asxl2*$^{-/-}$ mice compared with WT control mice. Furthermore, deletion of *Asxl2* significantly increased the GMP population with skewed differentiation favouring the monocytic/granulocytic lineage. Thus, *Asxl2* loss leads to an increased HSC self-renewal and a favourable myeloid differentiation. A recent study by Ye *et al.*[24] showed that skewed myeloid differentiation is a prerequisite for leukaemic stem cell formation and AML development. Consistently, skewed expansion of GMP population is associated with higher risks of leukaemic transformation in MDS patients[16,25]. The increased HSC self-renewal and a higher percentage of GMP in *Asxl2*-deficient mice probably facilitate leukaemogenesis.

Similar to *ASXL1* mutations, patients with *ASXL2* mutations had a higher incidence of relapse compared with *ASXL1/2* WT counterparts[13], suggesting a critical role of ASXL2 in driving disease progression. In a competitive transplantation assay, *Asxl2*$^{-/-}$ HSC/HPCs contributed to a transient decline following a steady ascend in the CD45.2 chimerism in recipients transplanted with *Asxl2*$^{-/-}$ BM cells. In addition, *Asxl2*$^{-/-}$ HSC/HPCs gave rise to a greater proportion of myeloid cells in the recipient BM than WT HSC/HPCs. Importantly, both *Asxl2*$^{+/-}$ and *Asxl2*$^{-/-}$ BM recipient mice had a shortened lifespan due to the development of myeloid malignancies. It is possible that the transplantation stress may promote the clonal evolution/ transformation of *Asxl2*-deficient HSC/HPCs. These competitive transplantation data suggest that *Asxl2*-loss-mediated haematological phenotype is likely to be cell autonomous. *ASXL2* mutations frequently occur in *t*(8; 21) AML and are associated with a higher risk of relapse[13]. Future studies to investigate the synergistic/additive effects of *Asxl2* loss plus AML1-ETO in leukaemogenesis are undergoing.

Epigenetic regulation is central to the maintenance of cell identity and disturbance of epigenetic regulation is known to promote tumourigenesis[26]. The GSEA results showed an increased expression of genes related to nucleosome assembly and chromatin assembly signatures in *Asxl2*$^{-/-}$ LK cells, revealing a possible mechanism by which ASXL2 regulates haematopoiesis. ASXL family proteins all contain a plant homeodomain domain, as many histone-modifying proteins do, which can bind and recruit epigenetic regulators to modify the

histones. As a result, modified histones affect the neighbouring gene transcription. The ASXL2 target genes related to nucleosome and chromatin assembly might be through ASXL2-mediated histone modification regulation.

Histone modifications, such as methylation and acetylation, are a key epigenetic mechanism that regulates gene expression. Our ChIP-seq analysis revealed that loss of *Asxl2* altered the levels of histone enhancer marks, including H3K27ac, H3K4me1 and H3K4me2. There was a substantial association between the changes in these histone modifications and gene expression levels. More importantly, we showed that H3K27ac genomic enrichment exhibited a higher degree of alteration in *Asxl2*$^{-/-}$ LK cells and had a greater impact on gene expression among the three histone modifications examined. The deregulated genes were enriched for myeloid and haematopoietic cell development, which is consistent with the haematopoietic phenotypes observed in *Asxl2*-deficient mice. Collectively, these data demonstrate a critical role for ASXL2 in the regulation of haematopoiesis via the maintenance of the histone modification status, in particular H3K27ac. Our IP result showed that ASXL2 indeed interacted with histone deacetylases HDAC1 and HDAC2, suggesting that ASXL2 involves in recruiting/stabilizing HDAC complex to modulate H3K27 acetylation levels. Further investigations to precisely define the mechanism of ASXL2-mediated HDAC activity in haematopoiesis are warranted.

Recently, Lai *et al.*[8] showed that deletion of *Asxl2* led to reduced global H3K27me3 levels in heart cells. Our study showed that ASXL2 also regulate gene expression via altering the locus-specific levels of H3K27ac, as well as H3K4me1/2 in LK cells. As an example, *Asxl2*$^{-/-}$ LK cells had significantly upregulated expression of *Prmd16*. ChIP-seq analysis confirmed the higher H3K27ac signal at the promoter region of *Prdm16*, suggesting an association between H3K27ac and *Prdm16* expression. Interestingly, increased expression of *PRDM16* had been reported in a subgroup of AML patients[20]. We and others have reported that ASXL1 is required for polycomb repressive complex 2 binding at target loci in LK cells[5–7]. Our results showed that loss of *Asxl2* was associated with alterations in the occupancy of putative enhancers marked by H3K4me1/2 and H3K27ac in the genome. We, however, cannot exclude a role of other regulatory mechanisms that contribute to the gene alteration, such as insulators.

Collectively, our results demonstrate that ASXL2 plays an important role in normal haematopoiesis and *Asxl2* loss in mice leads to myeloid malignancies. The *Asxl2* knockout mice present an ideal model for unveiling the mechanisms underlying the *Asxl2*-loss-mediated multiple-step pathogenesis of myeloid malignancies and for testing novel therapeutic agents for myeloid malignant patients with *ASXL2* mutations. ASXL1 and ASXL2 probably have both overlapping and unique functions in normal and malignant haematopoiesis. Future work is warranted to

**Figure 6 | ASXL2 maintains transcriptional programs through H3K27ac and H3K4me1/2.** (**a**) Scatter plots with kernel density estimation showing correlations between change in H3K27ac and H3K4me1/2, and changes in gene expression in WT and *Asxl2*$^{-/-}$ LK cells. Grey and red points represent all expressed genes and DEGs, respectively. Solid line, linear fit. Spearman's correlation coefficient *R*-values are shown, $P < 10^{-7}$ for all correlation tests. *X* and *Y* axes show the log$_2$ transformed fold change (log$_2$FC) for each histone enrichment and mRNA expression level, respectively. (**b**) Boxplot displaying enrichment difference in H3K27ac, H3K4me1 and H3K4me2 at increased (up) and decreased (down) enrichment regions between WT and *Asxl2*$^{-/-}$ LK cells. *Y* axis shows the log$_2$FC in histone modification. $**P < 0.001$ (unpaired Student *t*-test). (**c**) Line plots representing mRNA expression differences for genes with increased and/or decreased signals in H3K27ac, H3K4me1 and H3K4me2 between WT and *Asxl2*$^{-/-}$ LK cells. *X* axis and *Y* axis with log$_2$ scale show the rank of genes with differential signals according to the degree of expression change in decreasing order. (**d**) Heatmap representing KEGG pathway enriched by the genes with differentially enriched regions (DERs) of H3K27ac, H3K4me1 and H3K4me2. Row and column represent KEGG pathway and gene set, respectively. Colour scale indicates enrichment significance. The number in each cell shows the number of enriched genes in this pathway. Up and down represent genes with increased and decreased signals, respectively. (**e**) Representative genome browser tracks showing H3K27ac, H3K4me1 and H3K4me2 enrichment on promoter regions of *Prdm16* and *Fas*, as well as their fold change between WT and *Asxl2*$^{-/-}$ LK cells. The red box indicates DER.

decipher this critical question and unveil the roles of *ASXL1* and *ASXL2* mutations in the pathogenesis of myeloid malignancies.

## Methods

**Asxl2-null-deficient mouse model and reagents.** The targeting strategy of the $Asxl2^{-/-}$ mice was reported previously by Baskind *et al.*[15]. Briefly, the gene trap cassette was integrated into the first intron of *Asxl2*; the exact site of integration was 5,016 bp downstream of exon 1. All studies were conducted in accordance with the regulatory guidelines by the Institutional Animal Care and Use Committee at University of Miami Miller School of Medicine. Chemicals were obtained from Sigma (St Louis, MO) unless otherwise indicated.

**Phenotypic analyses of the haematologic system in mice.** PB was collected by retro-orbital bleeding and was subjected to an automated blood count (Hemavet System 950FS). For morphological and lineage differential analysis, PB smears were subjected to May–Grünwald–Giemsa staining. Morphological analysis of BM and spleen cells were performed on cytospins followed by May–Grünwald–Giemsa staining. For histopathology analyses, femurs were fixed in formaldehyde and demineralized in a solution of 10% EDTA for 1–2 weeks. The specimens and other soft tissues (spleens and livers) were dehydrated using ethanol and cleared in xylenes. The specimens were then embedded in melted paraffin and allowed to harden. Thin sections (4–5 μm) were cut and floated onto microscope slides. For routine assessment, slides were stained with haematoxylin and eosin. For MPO staining, the tissue was rehydrated followed by heat-induced epitope retrieval, peroxidase and serum blocking. Samples were then incubated with MPO antibody (R&D, MAB3174) overnight at 4 °C followed by staining with the biotinylated second antibody. Slides were visualized under a Nikon TE2000-S microscope. Images were taken by a QImaging camera and QCapture-Pro software (Fryer Company Inc.).

Flow cytometric analysis, cell sorting and HSC/HPC assay. Total white blood cells were obtained after lysis of PB with red cell lysis buffer. Single-cell suspensions from the BM, spleen, liver and PB were stained with panels of fluorochrome-conjugated antibodies (Supplementary Table 1). Flow cytometric analysis of HSC/HPCs was performed as previously described[6]. Dead cells were excluded by 4,6-diamidino-2-phenylindole staining. The analyses were performed using a BD FACS Canto II or LSR Fortessa flow cytometer. For cell apoptosis analysis, freshly isolated BM cells were stained with lineage/cKit antibodies and PE-Annexin V/7-AAD apoptotic kit (BD Biosciences) according to the protocol and analysed with the Lin$^+$ subpopulation and Lin$^-$cKit$^+$ (LK) subpopulation. For cell cycle analysis, BM cells were labelled with BrdU for 45 min *in vitro*, stained with surface markers, treated with DNase and finally stained with BrdU and 7-AAD. All data were analysed by FlowJo-V10 software. For LK cell selection, magnetic-activated cell sorting was applied. BM cells were first sorted with lineage depletion beads (Miltenyi Biotec, Bergisch Gladbach, Germany) and the lineage-negative cells were then sorted with cKit (CD117) beads. The purity of selected LK cells was routinely over 95%. For CFU assays, BM or spleen cells were plated in triplicate in methylcellulose medium (Methocult M3134, StemCell Technologies) supplemented with mouse (m) stem cell factor (mSCF, 100 ng ml$^{-1}$), human interleukin 6 (IL-6, 50 ng ml$^{-1}$), IL 3 (mIL-3, 5 ng ml$^{-1}$), erythropoietin (EPO, 4 U ml$^{-1}$), thrombopoietin (mTPO, 100 ng ml$^{-1}$) and mouse granulocyte–macrophage colony-stimulating factor (10 ng ml$^{-1}$, Peprotech), and scored on day 7 of the cultures[27].

**Paired daughter cell assay.** To examine the frequency of HSCs self-renewal and differentiation capability, we performed paired daughter cell assays[28]. Single CD34$^-$ LSK cells from BM of WT and $Asxl2^{-/-}$ mice were clone sorted into 96-well plates. The cells were maintained in RPMI1640 media supplemented with mSCF (100 ng ml$^{-1}$) and mTPO (50 ng ml$^{-1}$). After the first cell division, the two daughter cells were separated, one per well for an additional 12 days in the media supplemented with mSCF, mTPO, mEPO, mIL-3 and granulocyte colony-stimulating factor (mG-CSF). The self-renewal and differentiation capabilities of cultured CD34$^-$ LSK cells were determined by morphological analyses microscopically following haematoxylin and eosin staining. A total of 288 single cells (three 96-well plates) was analysed to calculate the percentage of symmetric/asymmetric cell divisions.

**Single-cell colony assay.** To explore the proliferative potential of single LT-HSC, we performed the single-cell colony assay as previously described[29]. Single CD34$^-$ LSK cells were sorted into 96-well plates with 10% fetal bovine serum, 1% BSA, 2 mM L-glutamine and $5 \times 10^5$ M 2-β-mercaptoethanol in α-MEM, supplemented with cytokines cocktails, including mSCF (20 ng ml$^{-1}$), mTPO (20 ng ml$^{-1}$), mIL-3 (10 ng ml$^{-1}$) and mEPO (4 U ml$^{-1}$). The colonies were scored after 7 days and the progenitors of individual colonies were collected and replated in a new 96-well plate. Cytospin slides were prepared from individual colonies followed by May–Grünwald–Giemsa staining. Antibodies used for the single cell sorting were shown in Supplementary Table 1.

**HPP-CFC assay.** Bone marrow mononuclear cells ($1 \times 10^5$) from WT, $Asxl2^{+/-}$ or $Asxl2^{-/-}$ mice were cultured in triplicate in 35-mm plates using soft agar for the growth of HPP CFCs in the presence of mSCF (100 ng ml$^{-1}$), human IL-6

(50 ng ml$^{-1}$), hFlt3L (10 ng ml$^{-1}$), mIL-3 (10 ng ml$^{-1}$), mouse granulocyte–macrophage colony-stimulating factor (10 ng ml$^{-1}$) and mG-CSF (10 ng ml$^{-1}$). Cultures were incubated at 5% $CO_2$, 5% $O_2$ and scored by indirect microscopy on day 14 for HPP-CFCs.

**Competitive repopulation assay.** The competitive repopulation assay was performed by transplanting BM cells ($1 \times 10^6$, CD45.2) from WT, $Asxl2^{+/-}$ or $Asxl2^{-/-}$ mice plus the competitor BM cells ($1 \times 10^6$, CD45.1) from B6.SJL mice, into lethally irradiated (9.5 Gy) recipients (CD45.1) by tail vein injection. Mice were monitored daily for signs of disease development.

**Lentiviral transduction.** The RNAi Consortium (TRC)-based *Prdm16* short hairpin RNA lentiviral vector (GE Healthcare Dharmacon, Inc.) were transfected into HEK 293T cells (ATCC) to package viruses. The supernatant was collected 48 h after transfection. The FL cells from WT and $Asxl2^{-/-}$ embryos (E12.5) were transduced with the packaged viruses and selected with puromycin. FL cells were collected into different dishes for colony-forming assays and liquid cultures.

**qPCR and RNA-seq analysis.** Total RNA was isolated from BM LK cells of each mouse genotype and treated with RNase-free DNase to remove contaminating genomic DNA. First-strand complementary DNA was synthesized. qPCR was performed using Fast SYBR Green master mix (Applied Biosystems). PCR amplifications were performed in triplicate for each gene of interest along with parallel measurements of *Actin* cDNA (internal control). To confirm specific amplification of the desired PCR product, melting curves were analysed and PCR products were separated on a 2% agarose gel. The primers used for the amplification of each gene are shown in Supplementary Table 2.

Total RNA was isolated from mouse $Asxl2^{-/-}$ and WT LK cells following standard protocol with TRIZol reagent (Life Technologies) followed by RNA library preparation with the Illumina TruSeq strand-specific mRNA sample preparation system. All RNA-seq libraries were sequenced with a read length of single-end 75 bp using the Illumina NextSeq 500 and final of over 45 million reads per sample.

RNA-seq reads were aligned to the mouse genome reference sequence (GRCm38/mm10) and RefSeq annotation (Release 23 May 2014) obtained from UCSC using TopHat (v2.0.9)[30] with a parameter of --max-multihits = 1. FastQC (v0.10.1) (http://www.bioinformatics.babraham.ac.uk/projects/fastq) and RseQC (v2.3.8) software[31] were used to check the quality of the raw reads. The quality score of each read reached an average of 25. The rates of overall mapped reads reached approximately 95%. Cufflink software (v2.0.2)[30] was used to quantify the gene expression values by calculating the FPKM value (the fragment per kilobase of transcript per million fragments mapped). Cuffdiff[30] was used to detect the DEGs and DEGs with a cutoff of $P < 0.05$ were selected for further pathway analyses. Expression profiles for all genes including DEGs are shown in Supplementary Dataset 1. GSEA analysis[32] was performed with gene signatures in GESA/MSigDB v5.1, including KEGG pathway signatures and Gene Ontology signatures. Enriched gene sets or pathways were selected using a cutoff of FDR < 0.25.

**Western blotting and IP assays.** IP was performed[33] using nuclear fraction buffer and antibodies (ASXL2 at a dilution of 1:500, Santa Cruz, sc-169972; HDAC1 at a dilution of 1:500, ThermoFisher Scientific, PA1-860; and HDAC2 at a dilution of 1:2,000, PA1-861). After washing with IP buffer (20 mM Tris-HCl, pH 7.5, 150 mM NaCl, 1% Triton X-100, 5 mM EDTA, 2 mM sodium orthovanadate, 1 mM phenylmethylsulfonyl fluoride, 2 mM NaF and protease inhibitor cocktail (Roche)) for four times, the associated proteins were collected for western blot analysis[34]. The histone modifications and protein levels were determined using antibodies at dilutions of 1:800 against H3K4me3 (Abcam, ab8580) and H3K27me3 (ab6002).

**Chromatin immunoprecipitation.** LK cells of each mouse genotype were fixed with 1% formaldehyde for 15 min and quenched with 0.125 M glycine. Chromatin was isolated by the addition of lysis buffer, followed by shearing with Bioruptor Pico with water cooler (Diagenode, Seraing, Belgium). The DNA was sheared to an average length of 300–500 bp. Genomic DNA regions of interest were isolated using antibodies against H3K27ac (Diagenode, C15410196), H3K4me1 (C15410194) and H3K4me2 (Abcam, ab32356). Complexes were washed, eluted from the beads with SDS buffer and subjected to RNase and proteinase K treatment. Crosslinks were reversed by incubation overnight at 65 °C and ChIP DNA was purified by phenol–chloroform extraction and ethanol precipitation. After the measurement of DNA concentration with Qubit3.0, the libraries were prepared and sequenced on an Illumina NextSeq 500.

**ChIP sequencing and data analysis.** ChIP-seq libraries for H3K27ac, H3K4me1 and H3K4me2 were prepared and sequenced with a read length of single-end 75 bp on NextSeq 500 and a final of over 45 million reads per sample. The sequence reads were aligned to the mouse genome (GRCm38/mm10) using the bowtie2 (v2.1.0) algorithm with default settings. The rates of overall mapped reads were at least 95%. Reads that aligned to the same position and strand were only counted once to

avoid potential PCR bias. Finally, 40–70% of total reads were kept for further analysis. Enrichment peaks of histone modifications were called using the MACS (v14)[35] with parameters of --fe-min = 1 --fe-max = 40 and filtered according to FDR ≤ 0.01, mean signal ≥ 0.02 and enrichment fold ≥ 2.5. The resulting genomic signals in wiggle files were visualized using UCSC genome browser (http://genome.ucsc.edu/). MAnorm[36] was used to identify differential enrichment regions for each histone modification between $Asxl2^{-/-}$ and WT LK cells, with $P \leq 0.01$ and read count after normalization to reads per million (RPM) ≥ 32. Peaks, binding sites and differential enrichment regions were assigned to its nearest genes within 100 kb using RefSeq annotation (Release 23 May 2014) from UCSC. Peaks overlapping with the regions from upstream 2 kb to downstream 2 kb of TSS (TSS ± 2 kb) were allocated to the promoter. Peaks overlapping with gene body rather than promoter were allocated to gene body. The remaining intergenic peaks were assigned to both the nearest 5′- and 3′-genes within its 100 kb. If no gene was found within 100 kb of an intergenic peak, it was not assigned to any gene. The binding sites and differential enriched regions in H3K27ac and H3K4me1/2 are shown in Supplementary Data set 2, respectively. Peak overlapping and merging were done with multovl (v1.2)[37]. ChIP-seq read counts were normalized to reads per kilobase per million reads and combined into one matrix with deeptools[38] (http://deeptools.ie-freiburg.mpg.de/) for heatmap visualization. Given 80% of histone modification peaks were located in regions ranging from upstream 10 kb to downstream 10 kb of gene body (gene ± 10 Kb), the ChIP-seq reads within Gene ± 10 kb were counted using cisgenome (v2)[39] with a parameter of −e200 and normalized to RPM for integrational analysis of ChIP-seq and gene expression data[39]. For each gene, enrichment fold was defined as $\log_2$ transformed RPM in ChIP divided by RPM in input, $\log_2$-(ChIP/input) and revised to zero when it was smaller than zero. This enrichment fold matrix was used to do GESA analysis of DEGs (http://software.broadinstitute.org/gsea)[32], to examine whether DEGs have differential enrichment signals of H3K27ac and H3K4me1/2. All figures were visualized in R using ggplot2 and pheatmap and GraphPad (v5.0).

**Pathway and biological functional enrichment analyses.** The identified DEGs and genes with the differential signals of H3K27ac, H3K4me1 or H3K4me2 in $Asxl2^{-/-}$ LK cells compared with WT cells were used for the pathway and biological function enrichment analyses with the DAVID Bioinformatics Resources v6.7 (https://david.ncifcrf.gov/)[40] and IPA. Enriched KEGG pathways ($P < 0.05$ and FDR ≤ 0.25) using DAVID were shown in heatmap. Enriched terms with the association to diseases and biological functions ($P < 0.05$ and $|z\text{-score}| \geq 1.5$) using IPA were shown in Enrichment Map for function clusters[41].

**Statistical analysis.** Differences between experimental groups were determined by the Student's $t$-test or analysis of variance followed by Newman–Keuls multiple comparison tests as appropriate. $P < 0.05$ is considered significant. For the in vivo experiment, sample size chosen was based on the generalized linear model with Bonferroni multiple comparison adjustments, with the proposed sample size of at least five mice per group/genotype. Animals were randomly assigned to each study. For all in vitro experiments, at least three independent experiments with more than three biological replicates for each condition/genotype were performed to ensure adequate power.

**Data availability.** Raw data generated from RNA-seq and ChIP-seq have been deposited in the Genome Sequence Archive (http://gsa.big.ac.cn/) in BIG Data Center[42] under accession number PRJCA000259. All other remaining data are available within the Article and Supplementary Files, or available from the authors upon request.

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

## Acknowledgements

This work was supported by grants from the National Institutes of Health (CA172408 and CA185751 to F.-C.Y. and M.X., and HL112294 to M.X.), the National Key Basic Research Program of China (2014CB542001 to Q.-F.W.), the External Cooperation Program of BIC, Chinese Academy of Sciences (grant 153F11KYSB20150013 to Q.-F.W.), the National Natural Science Foundation of China (81470340 to F.H.) and the Youth Innovation Promotion Association of Chinese Academy of Sciences (F.H.). We thank Zehui Xu, Yuguang Ban, Jin Li and Ehsan Hajiramezanali for the statistical data analysis. We also thank the services provided by the Satellite Histological Core, Flow Cytometry Core, Oncogenomics Core and Biostatistics and Bioinformatics Core Facilities of Sylvester Comprehensive Cancer Center, University of Miami Miller School of Medicine.

## Author contributions

F.-C.Y. conceived the project. J.L., F.H., P.Z., Q.-F.W., M.X. and F.-C.Y. designed the study. J.L., P.Z., S.C., H.S., Y.G., H.Y., N.M., S.G. and Z.L. performed the cellular and molecular experiments. F.H., P.Z., J.L., Y.S., Z.G., S.W., X.C., P.Y., Q.-F.W. and F.-C.Y. performed the RNA-seq and ChIP-seq analysis. J.L., S.C., M.X. and F.-C.Y. reviewed the blood smears and histopathological sections. J.L., P.Z., Y.Z., L.W., M.L., Q.W., S.N., M.X. and F.-C.Y. discussed and analysed the data. J.L., F.H., P.Z., Q.-F.W., M.X. and F.-C.Y. wrote the manuscript. All authors reviewed and approved the manuscript.

## Additional information

**Competing interests:** The authors declare no competing financial interests.

