## [Peer Review File · Nature Communications]

Reviewers' comments:

Reviewer #1 (Remarks to the Author): Expert in leukaemia and epigenetics

In this study, the authors showed that deletion of *Asxl2* in mice leads to the development of MDS-like disease. *Asxl2*^{-/-} mice had an increased bone marrow HSPCs and showed myeloid-biased differentiation compared to wild-type controls. Recipients transplanted with *Asxl2*^{-/-} and *Asxl2*^{+/-} bone marrow cells showed shortened survival because of the development of MDS or AML. Deletion of *Asxl2* altered the expression of genes critical for apoptosis and HSC self-renewal and differentiation in HPCs. The altered gene expression was associated with the dysregulated enhancer markers. Based on these findings, the authors concluded that ASXL2 functions as a tumor suppressor to maintain normal HSC function.

The study clearly uncovered the role of *Asxl2* in hematopoiesis and leukemogenesis. However, the authors failed to uncover the relationship between *Asxl2* and t(8;21) fusion gene that are observed in human AML. Furthermore, the authors did not define the genome-wide localization of *Asxl2* or biochemical characteristics of the *Asxl2* complex, leaving the impact of *Asxl2* on transcription and epigenome obscure. Additionally, I have several points as follows.

Major:

1. *Asxl2* KO HSPCs shows growth advantage in culture and are less apoptotic in vivo. These phenotypes of *Asxl2* KO HSPCs in gene trap mice are not compatible with MDS. In addition, ASXL2 mutations are rare in MDS. What is known about ASXL2 in MDS? Is it downregulated in MDS?
2. In contrast to the growth advantage of *Asxl2* KO BM HSPCs in culture, it shows lower repopulating capacity in competitive repopulation assays. How do the authors explain this discrepancy? Why *Asxl2*^{+/-} does not show impaired repopulation?
3. Do the straight *Asxl2* KO also develop leukemia?
4. In competitive repopulation assays, some of the mice show MDS-like phenotype and others develop AML. Does this indicate that AML develop from the MDS clone? Do all the recipient mice show MDS phenotype at the early time points? Please present the frequency of the *Asxl2*^{+/-} and *-/-* mice developing MDS. Do the MDS mice show increased apoptosis not only in differentiated cells but also in HSPCs?
5. In Figure 6, without *Asxl2* ChIP-seq data or biochemical data of *Asxl2* complex, we cannot evaluate the association of *Asxl2* function with enhancer histone modifications. The notion that *Asxl2* regulates enhancer H3K27ac has not been supported by any direct experimental data in this study. Is there any molecular mechanism known about this?

Minor:

1. Although it has been already reported, please briefly describe the targeting strategy of *Asxl2* in mice. Which portion of *Asxl2* is deleted by gene trap? Are they null mice?
2. Is there any specific differentiation block in lymphocytes? Please show L-MPP, CLPs, B-cell progenitor fractions in BM and thymocytes in the thymus.
3. In Figure 2, please show the total BM cell numbers.
4. In the abstract, the authors claim that AML1-ETO9a cooperated with *Asxl2* deficiency to promote HSC self-renewal. But, from this experiment (Supplemental Figure 3d), we cannot judge the HSC self-renewal. This is just a progenitor assay.

Reviewer #2 (Remarks to the Author): Expert in haematopoiesis

In this manuscript, Li and colleagues hypothesized that ASXL2 might have a role in hematopoiesis, starting from the observation that ASXL2 is found mutated in AML with the t(8:21) translocation. In their model, the total deletion of ASXL2 leads to an increased HSC number and function and development of MDS. Moreover, they show that ASXL2 regulates gene expression through histone modification at the enhancers. This manuscript describes a new role of ASXL2 in hematopoiesis

and provides a better understand of hematopoiesis in both normal and leukemic contexts. Overall, the experiments were well performed and the results are interesting. The manuscript could be improved by addressing the comments below.

Major comments

- 1- In the Figure 3: The authors should further define the proliferative capacity (for example by BrdU incorporation) and cell cycle of ASXL2 deficient HSPCs
- 2- Supplementary figure 3d: Because ASXL2 mutations are found in AML-ETO AML, the authors tested the possible cooperation between ASXL2 loss and AML-ETO transformation. The data presented are quite sparse and this part could be removed from the manuscript, if no more detailed experiments are added, as the work featured is insufficient to be convincing and does not add a lot to the story.
- 3- Figure 4: After transplantation of the ASXL2^{-/-} BM in recipients, the mice die from what is described as a leukemia, with increased chimerism, high proportion of MAC1⁺ cells from the donor, 20% of ckit⁺ cells in the BM, splenomegaly, anemia and invasion of blast cells in the BM and spleen. However, in case of Acute Myeloblastic leukemia (as stated at least in the abstract), one would expect to see the chimerism increase beyond 50%; indeed the leukemic cells would overtake the competitive population. It would also be interesting to have the WBC count in the blood and a blood smear to assess the invasion of the blast cells in the blood. Overall, more detailed FACS analysis of the populations in the different organs (BM, Spleen and blood) should be included to have a better idea of which type of leukemia the recipient mice develop.
- 4- Figure 5: It is not clear how the authors came to look at their 3 targets genes (5d). It does not look very convincing to choose, without providing a rationale for it, 3 genes from a list of 150. If the authors want to claim that these genes are important for the observed phenotype, experiments should be conducted to this end. For example, does Knockdown of PRMD16 rescue the phenotype (in a CFU assay)?
- 5- There is no mention of the cut-off used to determine the differentially expressed genes in term of fold change (FC)? Was it 2?
- 6- Figure 6: In figure 5, the authors mention 68 up and 92 downregulated genes, whereas there are around 1000 sites with changes in histone modification. Can the authors comment on what happens to the genes with changes in histone modification not associated with a difference in gene expression as they represent the majority of the genes observed?
- 7-Figure 6e: The authors write "Furthermore, ChIP-seq analysis confirmed the higher H3K27ac peak at the distal region of Prdm16": in the data presented in the figure, the difference is not obvious. The FC between the 2 signals should be indicated.
- 8-The authors wrote "the 1559 genes with differential H3K27ac signals have higher mRNA expression difference with a median FC of 1.332"; however, this way of representing data is confusing. One would expect to have genes with more H3K27ac to be upregulated, and with less H3K27ac to be downregulated. Is it the case? Terms such as "differential", "change" or "dysregulation" should be clarified.
- 9-The authors should provide the list of genes with the data for the RNA-seq and Chip-seq.
- 10- It would be important to perform ASXL2 ChIP-seq to correlate change in gene expression and histone modifications to ASXL2 binding.
- 11- Figure 6a: The R values don't seem very high; could the authors comment on this point? Is

there really a substantial correlation, especially for H3K27Ac?

12- It would be interesting to know if the effects observed in adult mice are also observed during development, in particular in the fetal liver of the ASXL2 ko mice.

Minor comments

- The manuscript may benefit from proofreading by a native English reader
- Figure 2d: in the figure legend it is indicated "quantification for both LSK and LKS- population" however only LSK is found in the figure
- Figure 5: The number of differentially expressed genes does not appear in the main text, only in the figure legend; for the sake of clarity, these numbers should be cited in the main text.
- Figure 5: Please correct the numbering of the figure (c is missing)
- Figure 5b: The IPA representation is too small (text too small)
- Line 106: The authors mention pancytopenia but increased neutrophil counts. Lymphocyte counts would be useful.
- Line 156: The authors should discuss the significance of the expanded GMP population in the MDS/AML context?
- Methods – line 407-410: More details or reference would be appreciated
- Line 415: Could the authors detail what FACS antibodies were used?
- Line 440: "The self-renewal ability and differentiation capability of these single CD34–LSK cells were determined by the potential of retaining multipotent lineage differentiation potential or lineage commitment": please clarify how (single cell colony assay)?

Reviewers' comments:

Reviewer #1:

Major comments:

1. *Asx12* KO HSPCs shows growth advantage in culture and are less apoptotic in vivo. These phenotypes of *Asx12* KO HSPCs in gene trap mice are not compatible with MDS. In addition, ASXL2 mutations are rare in MDS. What is known about ASXL2 in MDS? Is it downregulated in MDS?

Response: We agree with the reviewer that apoptosis in BM cells of MDS patients is elevated. We found a lower apoptotic rate in the LK cells of *Asx12*^{-/-} mice. However, there was a higher apoptotic rate in mature lineage positive cells in *Asx12*^{-/-} mice compared to WT controls. In response to the reviewer's question, we examined the levels of ASXL2 mRNA expression in the BM CD34⁺ cells of a cohort of MDS patients (n=18) and compared with that in CD34⁺ cord blood cells and G-CSF mobilized CD34⁺ cells. A lower expression of ASXL2 was observed in four out of 18 MDS patients examined. These data suggest that ASXL2 is down regulated in at least some of the MDS patients. Future study with a larger cohort of patients is necessary to true frequency of downregulated ASXL2 expression in MDS patients.

[FIGURE REDACTED]

2. In contrast to the growth advantage of *Asx12* KO BM HSPCs in culture, it shows lower repopulating capacity in competitive repopulation assays. How do the authors explain this discrepancy? Why *Asx12*^{+/-} did not show impaired repopulation?

Response: We agree with the reviewer that there is a discrepancy in the phenotypes on HSPCs between the *in vivo* and *in vitro* assays. Our *in vivo* phenotypic analysis revealed that loss of *Asx12* mainly affected LT-HSCs with less apoptosis. In contrast, *Asx12* loss increased the apoptosis in lineage committed and matured cell populations, which might cause the transient reduction of repopulating capacity of *Asx12* deficient BM cells at early time points after transplantation. However, after five months, *Asx12* null LT-HSCs expanded in the recipient mice, resulting in a higher reconstitution rate. Our data showed that *Asx12* haploinsufficient mice exhibited mild hematopoietic phenotypes, except lower survival rate in the *Asx12*^{+/-} BM transplanted recipients. These data may suggest a gene dosage dependent effect of *Asx12* in the hematopoiesis.

3. Do the straight *Asx12* KO also develop leukemia?

Response: We thank the reviewer for this great question. To address this question, we analyzed a cohort of aged *Asx12*^{-/-} mice and found that *Asx12*^{-/-} mice around 12 months of age did develop leukemia as evidenced by high WBC counts, increased proportion of Gr1⁺/Mac1⁺, GMP in the BM, splenomegaly and hepatomegaly, as well as white femurs. LT-HSCs in the

BM were also much higher in these aged *Asx12*^{-/-} mice than that in WT controls. These data are now added as Supplementary Fig. 4a-e in the revised manuscript on page 9.

4. In competitive repopulation assays, some of the mice show MDS-like phenotype and others develop AML. Does this indicate that AML develop from the MDS clone? Do all the recipient mice show MDS phenotype at the early time points? Please present the frequency of the *Asx12*^{+/-} and *Asx12*^{-/-} mice developing MDS. Do the MDS mice show increased apoptosis not only in differentiated cells but also in HSPCs?

Response: MDS-like phenotypes can be seen in most of the *Asx12*^{+/-} or *Asx12*^{-/-} BM recipient mice before leukemic transformation. It is likely that transformation of leukemia in these recipients were evolved from the MDS clone. We presented these data in Supplementary Table 3 to summarize the frequency of MDS and AML post transplantation of *Asx12*^{+/-} or *Asx12*^{-/-} BM cells. Flow cytometric analysis revealed an increased percentage of apoptotic cells in Lin⁺ BM cells of MDS mice and a lower percentage of apoptotic cells in LK cells. These data are now added as Supplementary Fig. 8c in the revised manuscript.

5. In Figure 6, without ASXL2 ChIP-seq data or biochemical data of ASXL2 complex, we cannot evaluate the association of ASXL2 function with enhancer histone modifications. The notion that ASXL2 regulates enhancer H3K27ac has not been supported by any direct experimental data in this study. Is there any molecular mechanism known about this?

Response: We thank the reviewer pointed out an important question. We performed ASXL2 ChIP-seq assay to determine ASXL2 binding sites on the genome using WT LK cells. ChIP-seq analyses revealed that a 12% of ASXL2 peaks was located at promoter regions, and the rest of ASXL2 peaks were at gene body (41%) and intergenic regions (47%). GO/KEGG analysis revealed that ASXL2 binding sites were enriched at genes that are associated with cell differentiation, positive regulation of transcription from RNA polymerase II promoters, etc. These data are now shown as Supplementary Fig. 10b,c and on page 14 in the revised manuscript. Of note, similar to Dr. Abdel-Wahab's group, we only detected a low number of ASXL2 peaks, which might due to the low affinity of the ASXL2 antibody or relative low expression of ASXL2. Generation of high quality ASXL2 antibody is required for confirming these observations. In addition, two histone deacetylation-related proteins, HDAC1 and HDAC2 were identified to associate with ASXL2 by immunoprecipitation (IP) and western blot, implying the potential of involvement by ASXL2 in the regulation of H3K27 deacetylation. We recognize that further studies to elucidate the mechanism of ASXL2 in regulating HDAC activity and H3K27 acetylation is warranted. These data are now shown as Supplementary Fig. 13a-c and on page 16 in the revised manuscript.

Minor:

1. Although it has been already reported, please briefly describe the targeting strategy of *Asx12* in mice. Which portion of *Asx12* is deleted by gene trap? Are they null mice?

Response: We apologize for missing this information. The targeting strategy of the *Asx12*^{-/-} mice is now briefly added to the revised manuscript in the sections of Results on page 6 and Methods on page 21. Briefly, the gene trap cassette was integrated within the first intron of

Asx12, the exact site of integration is at 5016 bp downstream of exon 1. The cassette contains a polyA signal at the 3' end, interrupting endogenous splicing and causing translation to stop. Thus, the *Asx12*⁻ allele encodes a fusion protein containing the first 19 amino acids of *Asx12* joined to the β-gal reporter. This fusion protein is likely functionally null because it is missing all the conserved domains of wild-type ASXL2. Success targeting/deletion of *Asx12* gene transcription and therefore protein in hematopoietic cells are shown by PCR, qPCR and western blot, respectively (Supplementary Fig. 1).

2. Is there any specific differentiation block in lymphocytes? Please show L-MPP, CLPs, B-cell progenitor fractions in BM and thymocytes in the thymus.

Response: We thank the reviewer pointed out this important question. Flow cytometric analysis of BM revealed that the LMPP and CLP populations in *Asx12*^{-/-} mice were lower than those of WT controls. Furthermore, the early stage of B cells (pre-B1 and pre-B2) was decreased in the bone marrow of *Asx12*^{-/-} mice. These data are now added as Supplementary Fig. 3a,b. To investigate the T cell populations in the thymus *in vivo*, we also performed flow cytometric analysis using antibodies against CD25, CD44, CD4 and CD8 to distinguish the T cell populations as DN1 (CD25⁻CD44⁺CD4⁻CD8⁻), DN2 (CD44⁺CD25⁺CD4⁻CD8⁻), DN3 (CD44⁻CD25⁺CD4⁻CD8⁻), or DN4 (CD44⁻CD25⁻CD4⁻CD8⁻). The DN1, DN4, CD4⁺ and CD8⁺ populations were higher in *Asx12*^{-/-} thymus compared to WT controls. In contrast, DN2, DN3 and CD4⁺/CD8⁺ population were lower in *Asx12*^{-/-} thymus compared to WT controls. These data suggest that deletion of *Asx12* also interrupted lymphocyte development. These data are now shown as Supplementary Fig. 3c and on page 8 in the revised manuscript.

3. In Figure 2, please show the total BM cell numbers (age).

Response: As suggested by the reviewer, we now included the data showing the total BM cellularity in Supplementary Figure 1f in the revised version.

4. In the abstract, the authors claim that AML1-ETO9a cooperated with *Asx12* deficiency to promote HSC self-renewal. But, from this experiment (Supplemental Figure 3d), we cannot judge the HSC self-renewal. This is just a progenitor assay.

Response: We agree with the reviewer and performed additional experiments to further evaluate the cooperative effect between *Asx12*-loss and AML1-ETO. We firstly determined the effect of *Asx12* deletion and AML1-ETO expression on HSC self-renewal by transducing AML1/ETO 9a (AE9a) into fetal liver cells of WT and *Asx12*^{+/-} embryos. *Asx12*^{+/-} cells transduced with AE9a had a significantly higher replating potential than *Asx12*^{+/-} cells transduced with WT vector. We also genetically intercrossed *Asx12*^{+/-} mice with AML1/ETO knockin mice (*AML1-ETO-stop/+ Mx1-Cre*⁺, noted as AE) and generated *Asx12*^{+/-};AE mice. These mice were undergone three intraperitoneal injections with pl:pC. Three months after the pl:pC injection, we performed hematopoietic phenotypic analysis on PB and BM cells. The results showed that *Asx12*^{+/-};AE;*MxCre*⁺ mice had a higher proportion of LT-HSC in the BM, and higher frequencies of Mac1⁺ cells in the BM and the PB than those in *Asx12*^{+/-} or AE mice. The frequency of CFU-Cs was also significantly higher in the BM of *Asx12*^{+/-};AE mice (see data below). These data suggest that AML1-ETO cooperated with *Asx12* deficiency to promote

myeloid differentiation. We still have limited data to support the notion of *Asx12* deletion in cooperating with AML1-ETO in leukemia transformation. As suggested by reviewer 2, “The data presented are quite sparse and this part could be removed from the manuscript”, we now removed these data from the manuscript.

Reviewer #2

We are pleased that the reviewer feels “the experiments were well performed and the results are interesting”.

Major comments

1- In the Figure 3: The authors should further define the proliferative capacity (for example by BrdU incorporation) and cell cycle of *Asx12* deficient HSPCs.

Response: As suggested by the reviewer, we performed BrdU incorporation assays to determine cell cycle of *Asx12* deficient HSPCs. The result showed that *Asx12*^{+/-} and *Asx12*^{-/-} LK cells had an increased frequency of cells at S-phase compared to WT LK cells, suggesting a higher proliferation of *Asx12*^{+/-} and *Asx12*^{-/-} LK cells. These data are now provided in the revised manuscript as in Supplementary Fig. 6d.

2- Supplementary figure 3d: Because *ASXL2* mutations are found in *AML-ETO* AML, the authors tested the possible cooperation between *ASXL2* loss and *AML-ETO* transformation. The data presented are quite sparse and this part could be removed from the manuscript, if no more detailed experiments are added, as the work featured is insufficient to be convincing and does not add a lot to the story.

Response: We agree with the reviewer. As also suggested by the Reviewer 1, we further evaluated the cooperative effect between *Asx12*-loss and AML1-ETO *in vivo*, by genetically intercrossing *Asx12*^{+/-} mice with AML1/ETO knockin mice (*AML1-ETO-stop/+ Mx1-Cre*⁺, noted as *AE*) to generate *Asx12*^{+/-};*AE* mice. These mice were undergone three intraperitoneal injections with pl:pC over a six-day period of time. Three months after the pl:pC injection, we performed hematopoietic phenotypic analysis on PB and BM cells. The results showed that *Asx12*^{+/-};*AE*;*MxCre*⁺ mice had an increased proportion of LT-HSC in the BM, and higher frequencies of Mac1⁺ cells in the BM and PB compared to those in *Asx12*^{+/-} or *AE* mice. The frequency of CFU-Cs was also significantly higher in the BM of *Asx12*^{+/-};*AE* mice. These data now shown as below. We still feel there are limited data to support the notion of *Asx12* deletion in cooperating with AML1-ETO in leukemogenesis. As suggested by the reviewer, we removed the *Asx12*^{+/-};*AML1-ETO* data from the manuscript.

3- Figure 4: After transplantation of the *Asx12*^{-/-} BM in recipients, the mice die from what is described as a leukemia, with increased chimerism, high proportion of Mac1⁺ cells from the donor, 20% of ckit⁺ cells in the BM, splenomegaly, anemia and invasion of blast cells in the BM and spleen. However, in case of Acute Myeloblastic leukemia (as stated at least in the abstract), one would expect to see the chimerism increase beyond 50%; indeed the leukemic cells would overtake the competitive population. It would also be interesting to have the WBC count in the blood and a blood smear to assess the invasion of the blast cells in the blood. Overall, more detailed FACS analysis of the populations in the different organs (BM, Spleen and blood) should be included to have a better idea of which type of leukemia the recipient mice develop.

Response: According to the reviewer's comment, we now provided more detailed information with flow cytometric analyses. The data showed that leukemic mouse had higher percentage of Gr1⁺/Mac1⁺ cells in the BM, spleen and PB, as well as much higher percentage of cKit⁺/Mac1⁺ cells in BM and spleen compared to WT controls. These data are now added as Supplementary Fig. 7f in the revised manuscript. We agree with the reviewer and changed "AML" to myeloid leukemia in the "Abstract" to more precisely reflect the disease characteristics. Besides, a detailed FACS analysis of the populations in the different organs (BM, Spleen and blood) shown as Supplementary Fig. 7f-h in the revised manuscript. In addition, the WBC count in PB and a blood smear were shown as Supplementary Fig. 7d,e.

4- Figure 5: It is not clear how the authors came to look at their 3 target genes (5d). It does not look very convincing to choose, without providing a rationale for it, 3 genes from a list of 150. If the authors want to claim that these genes are important for the observed phenotype, experiments should be conducted to this end. For example, does Knockdown of *Prdm16* rescue the phenotype (in a CFU assay)?

Response: We choose these three target genes based on their known roles in apoptosis (for *Fas* and *Casp3*) or HSC self-renewal regulation (for *Prdm16*). Given the fact that *Prdm16* is required for hematopoiesis and maintenance of HSPCs self-renewal and *Prdm16* is higher expressed in AML, as suggested by the reviewer, we knocked down *Prdm16* using lentiviral-system in the fetal liver LK cells from WT and *Asx12*^{-/-} embryos. *Prdm16*-KD in *Asx12*^{-/-} fetal liver cells significantly decreased the percentage of LSK cells after seven days of culture as well as the frequency of CFU-C. These data are now shown as Supplementary Fig. 9b,c in the revised manuscript.

5- There is no mention of the cut-off used to determine the differentially expressed genes in term of fold change (FC)? Was it 2?

Response: We thank the reviewer for this comment. We identified a set of 565 differentially expressed genes (DEGs) in *Asx12*^{-/-} LK cells ($P < 0.05$), in which 160 DEGs ($P < 0.05$, FDR < 0.25 and fold change ≥ 1.5) are shown in Figure 5a. According to the reviewer's comment, we provided this information in the sections of Results and Figure legend in the revised manuscript on page 13.

6- Figure 6: In figure 5, the authors mention 68 up and 92 downregulated genes, whereas there are around 1000 sites with changes in histone modification. Can the authors comment on what happens to the genes with changes in histone modification not associated with a difference in gene expression as they represent the majority of the genes observed?

Response: We appreciate the reviewer's comment. There are a total of ~600 and ~900 genes with increased and decreased signals in H3K27ac, respectively. Overall, these 600 genes with increased histone signals were concordantly upregulated in *Asx2^{-/-}* LK cells compared to WT cells ($P < 0.01$ and $FDR < 0.01$), as shown by Gene Set Enrichment Analysis (GSEA) analysis. 40% (~260) were identified to be the leading-edge subset that contributes most to statistical significance of this analysis. The leading-edge subset identified with GSEA analysis is the core that accounts for the gene set's enrichment signal (shown in the following plot). In contrast, 68 upregulated genes mentioned were identified with Cuffdiff2 analysis at single gene level with $P < 0.05$, $FDR < 0.25$ and fold change ≥ 1.5 . These results suggest that a significant number (~260) of genes with increased levels of H3K27ac modification were upregulated, and a smaller set (68) of genes meet a more stringent criteria as upregulated. Similarly, these 900 genes with decreased histone signals were concordantly downregulated ($P < 0.01$ and $FDR < 0.01$), in which approximately 30% were identified to be the leading-edge subset that contributes most to statistical significance. 92 downregulated genes mentioned were identified with Cuffdiff2 analysis at single gene level with $P < 0.05$, $FDR < 0.25$ and fold change ≥ 1.5 . Similar analyses showed that genes with increased and decreased signals in H3K4me1/2 were concordantly upregulated and downregulated, respectively. To address the reviewer's question, we now provided the new data in Supplementary Fig. 11a.

7-Figure 6e: The authors write "Furthermore, ChIP-seq analysis confirmed the higher H3K27ac peak at the distal region of *Prdm16*": in the data presented in the figure, the difference is not obvious. The FC between the 2 signals should be indicated.

Response: We thank the reviewer for this suggestion. We examined the differences at single peak level, and identified several peaks with increased histone signals in this combined region. Among them, the peak in the promoter region of *Prdm16* had a greatest increase of three histone modifications (2.11, 1.81 and 1.68 for fold changes of H3K27ac, H3K4me1 and H3K4me2, respectively). Accordingly, we indicated this most prominent peak with fold changes between the histone signals in WT and *Asx2^{-/-}* cells in the revised Fig. 6e. We also re-worded the description in the Figure legends.

8-The authors wrote "the 1559 genes with differential H3k27ac signals have higher mRNA expression difference with a median FC of 1.332"; however, this way of representing data is confusing. One would expect to have genes with more H3k27ac to be upregulated, and with less H3k27ac to be downregulated. Is it the case? Terms such as "differential", "change" or "dysregulation" should be clarified.

Response: We agree with the reviewer's comment. Accordingly, we divided the 1559 genes with differential H3K27ac signals into the genes with increased H3K27ac signal and the genes with decreased H3K27ac signal. Overall, these genes with increased signals in H3K27ac, H3K4me1 or H3K4me2 were concordantly upregulated, and the genes with decreased signals

in H3K27ac, H3K4me1 or H3K4me2 were concordantly downregulated, in *Asxl2*^{-/-} LK cells compared to WT cells, as shown by Gene Set Enrichment Analysis (GSEA) analysis ($P < 0.01$ and $FDR < 0.01$). These new data are now shown as Fig. 6b,c and Supplementary Fig. 11a in the revised manuscript.

9-The authors should provide the list of genes with the data for the RNA-seq and Chip-seq.

Response: According to the reviewer's suggestion, we now provided this data as Supplementary Table 4 and Supplementary Table 6 to show the list of genes with both RNA-seq and ChIP-seq data.

10- It would be important to perform ASXL2 ChIP-seq to correlate change in gene expression and histone modifications to ASXL2 binding.

Response: We thank the reviewer pointed out an important question. We performed ASXL2 ChIP-seq assay to determine ASXL2 binding sites on the genome using WT LK cells. ChIP-seq analyses revealed that a 12% of ASXL2 peaks was located at promoter regions, and the rest of ASXL2 peaks were at gene body (41%) and intergenic regions (47%). GO/KEGG analysis revealed that ASXL2 binding sites were enriched at genes that are associated with cell differentiation, positive regulation of transcription from RNA polymerase II promoters, etc. These data are now shown as Supplementary Fig. 10b,c and on page 14 in the revised manuscript. Of note, similar to Dr. Abdel-Wahab's group, we only detected a low number of ASXL2 peaks, which might due to the low affinity of the ASXL2 antibody or relative low expression of ASXL2. Generation of high quality ASXL2 antibody is required for confirming these observations.

11- Figure 6a: The R values don't seem very high; could the authors comment on this point? Is there really a substantial correlation, especially for H3K27Ac?

Response: We appreciate the reviewer's comment. Our original Fig. 5a showed correlations between the changes in three histone modifications and mRNA expression difference in *Asxl2*^{-/-} cells compared to WT cells. Now, we also examined the correlation between histone modifications and mRNA expression in one condition. New analysis showed a high positive correlation with spearman correlation coefficient $R=0.68$ between H3K27ac enrichment and gene expression in WT and *Asxl2*^{-/-} LK cells ($P < 0.001$, Supplementary Fig. 10f), which is consistent with previous studies by others (reference^{21,22} in the revised manuscript). Similarly, R values ranging from 0.5 to 0.7 were identified for associations between H3K4me1/2 and mRNA expression in WT and *Asxl2*^{-/-} LK cells, respectively ($P < 0.001$). The relative low R values with about 0.25 may reflect the fact that change in mRNA expression is influenced by multiple factors, such as transcription factors, chromatin conformation, one or multiple long-range regulatory elements. These results are now shown as Supplementary Fig. 10f and on page 15 in the revised manuscript.

12- It would be interesting to know if the effects observed in adult mice are also observed during development, in particular in the fetal liver of the ASXL2 KO mice.

Response: The reviewer raised an important question. In response to the reviewer's suggestion, we performed flow cytometric analysis on the fetal liver cells of each of the mouse genotypes. An increased MPP and Gr1⁺/Mac1⁺ populations were observed in *Asx2*^{-/-} fetal liver cells, while the frequencies of LSK and GMP were comparable amongst the three genotypes. Consistently, the CFU-GEMM, CFU-GM and BFU-E were significantly higher in *Asx2*^{-/-} fetal liver cells than WT and *Asx2*^{+/-} fetal liver cells. These data are now provided as Supplementary Fig. 5a,b in the revised manuscript.

Minor comments

- The manuscript may benefit from proofreading by a native English reader.

Response: The current manuscript has been improved significantly after being edited by a native English reader.

- Figure 2d: in the figure legend it is indicated "quantification for both LSK and LKS- population" however only LSK is found in the figure.

Response: The data for quantification of LSK⁻ is shown in Supplementary Fig.3d in the revised version.

- Figure 5: The number of differentially expressed genes does not appear in the main text, only in the figure legend; for the sake of clarity, these numbers should be cited in the main text.

Response: The information for the number of differentially expressed genes is now provided in the Result section on page 13 in the revised manuscript.

- Figure 5: Please correct the numbering of the figure (c is missing)

Response: Sorry for the missing information. The "c" is added to the Figure Legend of the revised manuscript.

- Figure 5b: The IPA representation is too small (text too small)

Response: The text for IPA representation is enlarged as suggested in the revised manuscript.

- Line 106: The authors mention pancytopenia but increased neutrophil counts. Lymphocyte counts would be useful.

Response: According to the reviewer's suggestion, we added the lymphocyte counts of the peripheral blood in the Fig. 1a of the revised manuscript.

- Line 156: The authors should discuss the significance of the expanded GMP population in the MDS/AML context?

Response: Britta Will *et al.* revealed that lower-risk MDS is characterized by expansion of phenotypic common myeloid progenitors (CMP). In contrast, higher-risk MDS have higher proportion of granulocyte/monocyte progenitors (GMP) and these patients have the higher risk of AML transformation. The higher proportion of GMP in *Asx2* deficient mice may associate with disease progression *in vivo*. We added the reference and this description to the result section on page 9 of the revised manuscript.

- Methods – line 407-410: More details or reference would be appreciated.

Response: As the reviewer suggested, we provided detailed information regarding the histological staining method on page 21, 22 in the revised manuscript.

- Line 415: Could the authors detail what FACS antibodies were used?

Response: As the reviewer suggested, the detailed information of FACS antibodies were provided in Supplementary Table 1 in the revised manuscript.

- Line 440: “The self-renewal ability and differentiation capability of these single CD34⁻LSK cells were determined by the potential of retaining multipotent lineage differentiation potential or lineage commitment”: please clarify how (single cell colony assay)?

Response: We now added a brief description on page 23 in the revised manuscript.

Reviewers' comments:

Reviewer #1 (Remarks to the Author):

The authors responded to most of my concerns precisely. However, I still have one concern about the types of myeloid malignancies in *Asx2* KO. Supplementary Fig 7f shows an increase in Gr-1-Mac-1+ monocytes in PB, BM and spleen. PB smear images in Supplementary Fig 7e are poor in quality and hard to interpret. Based on these findings, I wonder the KO mice developed CMML. Can the authors check this possibility? In addition, please present the number of PB monocytes in Figure 1a and provide better smear images in Supplementary Fig 7e.

Reviewer #2 (Remarks to the Author):

Overall, the authors addressed the comments satisfactorily. Please find a few minor comments relating to the rebuttal which would improve the quality of the paper:

a. Major comment 3/Supplementary Figure 7 c-e: It would be helpful to display WT and MDS blood films next to the leukemic blood films for comparison.

b. Major comment 4/Supplementary Figure 9b-c: Whilst the phenotype associated with the *Prdm16* knockdown is very striking, data showing the level of *Prdm16* expression in WT and *Asx12*^{-/-} Scrambled and *Prdm16*-shRNA cells should be included (qPCR of the mRNA and/or Western blot of the protein). (In particular, this may help explain the partial phenotype observed in WT knockdown cells).

c. Major comment 8/Figure 6c: The X-axis title of Figure 6c should be clearer, to indicate that the genes are ranked according to histone modification alterations.

d. Major comment 10/Supplementary Figure 10b-c: The authors have now performed ASXL2 ChIP Seq. However, as the authors state, only a relatively small number of peaks were detected, presumably due to the low affinity of the antibody used. In its current form, the ASXL2 ChIP Seq data may be misleading and therefore the authors should consider removing them. It would be better to follow up the story with data validated with a higher quality ASXL2 antibody.

Reviewers' comments:

Reviewer #1:

The authors responded to most of my concerns precisely. However, I still have one concern about the types of myeloid malignancies in *Asx2* KO. Supplementary Fig 7f shows an increase in Gr-1-Mac-1+ monocytes in PB, BM and spleen. PB smear images in Supplementary Fig 7e are poor in quality and hard to interpret. Based on these findings, I wonder the KO mice developed CMML. Can the authors check this possibility? In addition, please present the number of PB monocytes in Figure 1a and provide better smear images in Supplementary Fig 7e.

Response: We are pleased that the reviewer feels “The authors responded to most of my concerns precisely”. Regarding the concern about the types of myeloid malignancies, we summarized the absolute numbers of monocytes in the PB of *Asx2*^{-/-} mice. After careful analysis of these data, we diagnosed these mice as MDS-like disease. The number of PB monocytes is now added to Figure 1a, and high quality images of the smear are provided in Supplementary Fig 7e.

Reviewer #2:

We are pleased that the reviewer feels “the authors addressed the comments satisfactorily”.

Question a. Major comment 3/Supplementary Figure 7 c-e: It would be helpful to display WT and MDS blood films next to the leukemic blood films for comparison.

Response: As the reviewer suggested, the WT and MDS *Asx2*^{-/-} blood films are now presented next to the leukemic *Asx2*^{-/-} blood films for comparison in Supplementary Fig 7e of the revised manuscript.

b. Major comment 4/Supplementary Figure 9b-c: Whilst the phenotype associated with the *Prdm16* knockdown is very striking, data showing the level of *Prdm16* expression in WT and *Asx2*^{-/-} Scrambled and *Prdm16*-shRNA cells should be included (qPCR of the mRNA and/or Western blot of the protein). (In particular, this may help explain the partial phenotype observed in WT knockdown cells).

Response: We thank the reviewer for this important point. As suggested by the reviewer, the levels of *Prdm16* mRNA expression by qPCR in Scrambled and *Prdm16*-KD WT and *Asx2*^{-/-} cells are now presented in Supplementary Fig 9b.

c. Major comment 8/Figure 6c: The X-axis title of Figure 6c should be clearer, to indicate that the genes are ranked according to histone modification alterations.

Response: We thank the reviewer to point this out, and we are sorry for missing the information. The X-axis of Figure 6c indicates the genes with differentially enriched histone modifications that are ranked according to the degree of mRNA expression change. We have changed the label of X-axis for Figure 6c.

d. Major comment 10/Supplementary Figure 10b-c: The authors have now performed ASXL2 ChIP Seq. However, as the authors state, only a relatively small number of peaks were detected, presumably due to the low affinity of the antibody used. In its current form, the ASXL2 ChIP Seq data may be misleading and therefore the authors should consider removing them. It would be better to follow up the story with data validated with a higher quality ASXL2 antibody.

Response: We agree with the reviewer. As suggested, we have removed the ASXL2 ChIP-seq data in the revised manuscript.

REVIEWERS' COMMENTS:

Reviewer #2 (Remarks to the Author):

The authors have adequately addressed the reviewers comments.